# The synaptic ribbon is critical for sound encoding at high rates and with temporal precision

Philippe Jean[1,2,3,4†], David Lopez de la Morena[1,3,4†], Susann Michanski[2,4,5,6†], Lina María Jaime Tobón[1,2,3,4,7,8†], Rituparna Chakrabarti[2,3,4,5,6], Maria Magdalena Picher[1,2,4], Jakob Neef[1,2,4,7,8], SangYong Jung[1,4,9], Mehmet Gültas[10], Stephan Maxeiner[11], Andreas Neef[12*], Carolin Wichmann[2,4,5,6*], Nicola Strenzke[2,4,13*], Chad Grabner[1,4,7*], Tobias Moser[1,2,4,7,8,14*]

[1]Institute for Auditory Neuroscience and InnerEarLab, University Medical Center Göttingen, Göttingen, Germany; [2]Collaborative Research Center, University of Göttingen, Göttingen, Germany; [3]Göttingen Graduate School for Neurosciences and Molecular Biosciences, University of Göttingen, Göttingen, Germany; [4]InnerEarLab, Department of Otolaryngology, University Medical Center Göttingen, Göttingen, Germany; [5]Molecular Architecture of Synapses Group, Institute for Auditory Neuroscience, University Medical Center Göttingen, Göttingen, Germany; [6]Institute for Biostructural Imaging of Neurodegeneration, University Medical Center Göttingen, Göttingen, Germany; [7]Synaptic Nanophysiology Group, Max Planck Institute for Biophysical Chemistry, Göttingen, Germany; [8]Auditory Neuroscience Group, Max Planck Institute for Experimental Medicine, Göttingen, Germany; [9]Neuro Modulation and Neuro Circuitry Group, Singapore Bioimaging Consortium (SBIC), Biomedical Sciences Institutes, Singapore, Singapore; [10]Department of Breeding Informatics, Georg-August-University Göttingen, Göttingen, Germany; [11]Institute for Anatomy and Cell Biology, University of the Saarland, Homburg, Germany; [12]Bernstein Group Biophysics of Neural Computation, Max Planck Institute for Dynamics and Self-Organization, Göttingen, Germany; [13]Auditory Systems Physiology Group, Department of Otolaryngology, University Medical Center Göttingen, Göttingen, Germany; [14]Center for Nanoscale Microscopy and Molecular Physiology of the Brain, University Medical Center Göttingen, Göttingen, Germany

*For correspondence:
aneef@gwdg.de (AN);
carolin.wichmann@med.uni-goettingen.de (CW);
nicola.strenzke@med.uni-goettingen.de (NS);
chad.grabner@mpibpc.mpg.de (CG);
tmoser@gwdg.de (TM)

†These authors contributed equally to this work

Competing interests: The authors declare that no competing interests exist.

**Abstract** We studied the role of the synaptic ribbon for sound encoding at the synapses between inner hair cells (IHCs) and spiral ganglion neurons (SGNs) in mice lacking RIBEYE (RBE$^{KO/KO}$). Electron and immunofluorescence microscopy revealed a lack of synaptic ribbons and an assembly of several small active zones (AZs) at each synaptic contact. Spontaneous and sound-evoked firing rates of SGNs and their compound action potential were reduced, indicating impaired transmission at ribbonless IHC-SGN synapses. The temporal precision of sound encoding was impaired and the recovery of SGN-firing from adaptation indicated slowed synaptic vesicle (SV) replenishment. Activation of Ca$^{2+}$-channels was shifted to more depolarized potentials and exocytosis was reduced for weak depolarizations. Presynaptic Ca$^{2+}$-signals showed a broader spread, compatible with the altered Ca$^{2+}$-channel clustering observed by super-resolution immunofluorescence microscopy. We postulate that RIBEYE disruption is partially compensated by multi-AZ organization. The remaining synaptic deficit indicates ribbon function in SV-replenishment and Ca$^{2+}$-channel regulation.

DOI: https://doi.org/10.7554/eLife.29275.001

## Introduction

Encoding and processing of sensory information in the ear and the eye rely on ribbon synapses. Described in the 1960s as an electron dense structure tethering a halo of vesicles (*Sjostrand, 1958*; *Smith and Sjostrand, 1961*), the function of the synaptic ribbon has remained enigmatic despite decades of work (recent reviews in *Lagnado and Schmitz, 2015*; *Moser and Vogl, 2016*; *Safieddine et al., 2012*; *Wichmann and Moser, 2015*). Approaches to ribbon function included studies that employed natural variation of ribbon size or abundance during diurnal cycle or hibernation (*Hull et al., 2006*; *Mehta et al., 2013*), photoablation (*Mehta et al., 2013*; *Snellman et al., 2011*) and genetic manipulation (*Dick et al., 2003*; *Frank et al., 2010*; *Jing et al., 2013*; *Khimich et al., 2005*; *Lv et al., 2016*; *Maxeiner et al., 2016*; *Sheets et al., 2011*; *Van Epps et al., 2004*). Mutations initially focused on the presynaptic scaffold protein bassoon that is required for ribbon anchorage to the AZ (*Dick et al., 2003*; *Khimich et al., 2005*) via interaction with RIBEYE (*tom Dieck et al., 2005*). However, bassoon also exerts direct effects on AZ function (*Davydova et al., 2014*; *Hallermann et al., 2010*; *Mendoza Schulz et al., 2014*) and, hence, distinguishing direct effects of bassoon deletion and those caused by ribbon loss remained challenging (*Jing et al., 2013*).

RIBEYE-disruption turned out to be difficult: it is transcribed from the same gene as CtBP2, an essential transcription factor, disruption of which causes embryonic lethality (*Hildebrand and Soriano, 2002*). Complete abolition of RIBEYE was hard to achieve in zebrafish (*Lv et al., 2016*; *Van Epps et al., 2004*) given their duplicated genome. In fact, despite targeting both *ribeye* genes, RIBEYE immunofluorescence remained present in the retina and hair cells displayed 'ghost ribbons': structures recognized by a synaptic vesicle-halo but lacking electron density (*Lv et al., 2016*). Complete disruption of RIBEYE expression and lack of retinal ribbons were recently reported in a mouse knock-out of the RIBEYE-specific exon (*Maxeiner et al., 2016*). This study proved that RIBEYE is required for ribbon formation in the mammalian retina and the observed ribbon loss grossly impaired glutamate release from bipolar cell terminals. The key conclusion was that ribbons help to couple voltage-gated $Ca^{2+}$-channels to vesicular release sites to enable tight, so-called $Ca^{2+}$-nanodomain control of exocytosis (*Maxeiner et al., 2016*), that was previously reported for ribbon synapses of ear and eye (*Bartoletti et al., 2011*; *Brandt et al., 2005*; *Graydon et al., 2011*; *Jarsky et al., 2010*; *Johnson et al., 2017*; *Pangršič et al., 2015*; *Wong et al., 2014*). By employing the most specific, yet chronic, manipulation of the ribbon, this functional study on ribbonless retinal rod bipolar cells also confirmed that RIBEYE/the ribbon promotes a large complement of vesicular release sites. However, the electrophysiology was performed on rod bipolar cells while the molecular anatomy (immunofluorescence) focused on rod photoreceptors. Since the structure and function of ribbons formed at these two different cell types are distinct, a simple structure-function model was not easy to derive from this study. Moreover, the consequences of ribbon loss remained to be investigated at the systems level. Here, we studied the effects of RIBEYE-disruption on synaptic sound encoding in the cochlea. Combining assessments of the molecular anatomy from electron and fluorescence microscopy with cell and systems physiology, we revealed a role for the synaptic ribbon in organizing the topography of the IHC AZ, in $Ca^{2+}$-channel regulation and in vesicle replenishment. In summary, we demonstrate that the synaptic ribbon is important for sound encoding at high rates and with temporal precision at IHC synapses.

## Results

### Genetic disruption of RIBEYE transforms ribbon-type AZs of IHC synapses into synaptic contacts with multiple small ribbonless AZs

We first employed immunohistochemistry to study IHCs of 3-week-old RIBEYE knock-out mice (RBE[KO/KO]), in which the unique A-domain exon of RIBEYE was deleted by Cre-mediated excision (described in *Maxeiner et al., 2016*). Next to the A-domain, RIBEYE contains a B-domain that is largely identical to the transcription factor CtBP2, which is spared by the genetic manipulation and used as a target in immunohistochemistry of ribbons and nuclei (*Figure 1A,B*; *Khimich et al., 2005*).

**eLife digest** Our sense of hearing relies on our ears quickly and tirelessly processing information in a precise manner. Sounds cause vibrations in a part of the inner ear called the cochlea. Inside the cochlea, the vibrations move hair-like structures on sensory cells that translate these movements into electrical signals. These hair cells are connected to specialized nerve cells that relay the signals to the brain, which then interprets them as sounds.

Hair cells communicate with the specialized nerve cells via connections known as chemical synapses. This means that the electrical signals in the hair cell activate channel proteins that allow calcium ions to flow in. This in turn triggers membrane-bound packages called vesicles inside the hair cell to fuse with its surface membrane and release their contents to the outside. The contents, namely chemicals called neurotransmitters, then travels across the space between the cells, relaying the signal to the nerve cell.

The junctions between the hair cells and the nerve cells are more specifically known as ribbon synapses. This is because they have a ribbon-like structure that appears to tether a halo of vesicles close to the active zone where neurotransmitters are released. However, the exact role of this synaptic ribbon has remained mysterious despite decades of study.

The ribbon is mainly composed of a protein called Ribeye, and now Jean, Lopez de la Morena, Michanski, Jaime Toboń et al. show that mutant mice that lack this protein do not have any ribbons at their "ribbon synapses". Hair cells without synaptic ribbons are less able to timely and reliably send signals to the nerve cells, most likely because they cannot replenish the vesicles at the synapse quickly enough. Further analysis showed that the synaptic ribbon also helps to regulate the calcium channels at the synapse, which is important for linking the electrical signals in the hair cell to the release of the neurotransmitters.

Jean et al. also saw that hair cells without ribbons reorganize their synapses to form multiple active zones that could transfer neurotransmitter to the nerve cells. This could partially compensate for the loss of the ribbons, meaning the impact of their loss may have been underestimated. Future studies could explore this by eliminating the Ribeye protein only after the ribbon synapses are fully formed.

These findings may help scientists to better understand deafness and other hearing disorders in humans. They will also be of interest to neuroscientists who research synapses, hearing and other sensory processes.

DOI: https://doi.org/10.7554/eLife.29275.002

Synaptic ribbons of IHC afferent synapses were identified as presynaptic RIBEYE/CtBP2-immunofluorescent spots in wild-type (*Figure 1B*, RBE$^{WT/WT}$) and heterozygous (*Figure 1—figure supplement 1*, RBE$^{WT/KO}$) mice. Their number per IHC did not change in the heterozygous condition (15.5 ± 0.7, S.D. = 1.58; *n* = 50 cells, N = 4 for RBE$^{WT/KO}$ vs. 15.7 ± 1.1, S.D. = 2.19; *n* = 39 cells, N = 3 for RBE$^{WT/WT}$ at P21), while their intensity was significantly reduced (in arbitrary units: 3.4 ± 0.7, S.D. = 1.78; *n* = 600 spots for 40 cells, N = 3 in RBE$^{WT/KO}$ vs. 5.1 ± 1.1, S.D. = 2.23; *n* = 411 spots for 29 IHCs, N = 3 in RBE$^{WT/WT}$; p<0.0001, Mann-Whitney-Wilcoxon test; *Figure 1—figure supplement 1A,B*). RBE$^{KO/KO}$ IHCs lacked synaptic RIBEYE/CtBP2 immunofluorescence spots (*Figure 1B*), while immunolabeling of nuclear CtBP2 remained present, corroborating previous findings in the retina (*Maxeiner et al., 2016*). The number of afferent synapses per IHC was determined by the count of postsynaptic densities (PSDs) identified as PSD-95 immunofluorescent spots (*Figure 1C,D,E*) and was unchanged when RIBEYE was removed (13.7 ± 0.8, S.D. = 2.04; *n* = 56 cells, N = 4 in RBE$^{KO/KO}$ vs. 12.9 ± 0.6, S.D. = 2.13; *n* = 55 cells, N = 5 in RBE$^{WT/WT}$). Bassoon (*Figure 1C*) and RIM2 (*Figure 1D*), both presynaptic scaffold proteins (*Khimich et al., 2005*; *Jung et al., 2015a*), remained present at the ribbonless afferent synapses of RBE$^{KO/KO}$ IHCs (marked by PSD-95). The scaffold protein piccolino, the short isoform of piccolo (*Regus-Leidig et al., 2013*) that is present in cochlear and retinal ribbons (*Khimich et al., 2005*; *tom Dieck et al., 2005*; *Regus-Leidig et al., 2013*), was absent from afferent synapses of RBE$^{KO/KO}$ IHCs. However, piccolo immunofluorescence was present in the vicinity of afferent synapses likely marking the long form piccolo at the efferent presynaptic AZs (*Figure 1E*, see schematic in *Figure 1A*). The PSD areas were calculated by fitting a 2-

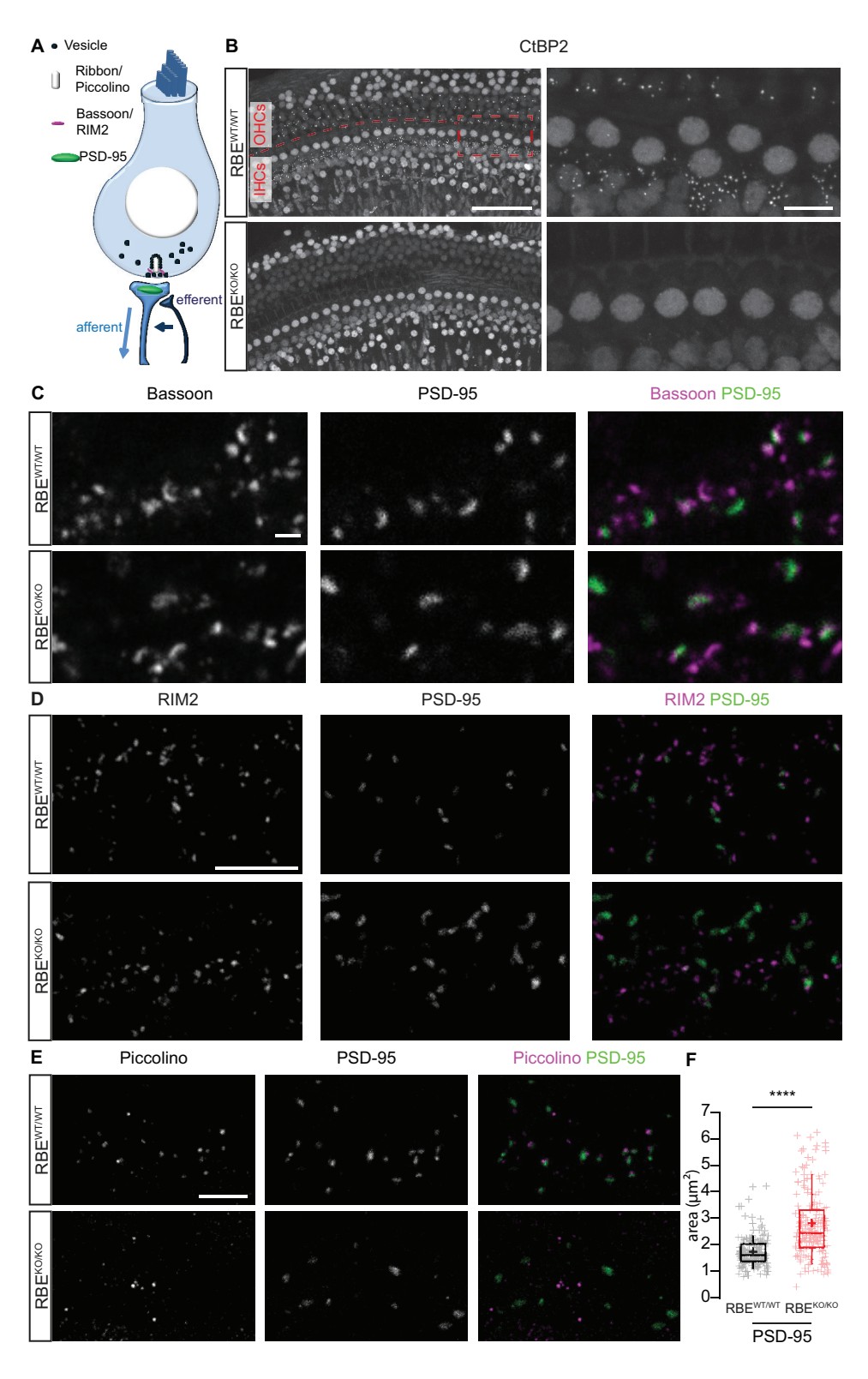

**Figure 1.** Loss of synaptic ribbons and piccolino from the AZs of RIBEYE-deficient IHCs. (**A**) Simplified schematic representation of an IHC with the afferent and efferent connectivities. (**B**) Maximal projection of confocal sections from organs of Corti immunolabeled for CtBP2 and RIBEYE, present in the nuclei and the ribbons, respectively. The RBE^(WT/WT) staining (top row) shows small puncta in the outer hair cell (OHC) and IHC rows representing the synaptic ribbons, which are completely absent in the RBE^(KO/KO) hair cells (bottom row). Scale bar = 50 μm. Zoom into the IHC row (right column),

*Figure 1 continued on next page*

*Figure 1 continued*

emphasizes the complete disappearance of CtBP2-labeling at the basolateral part of RBE[KO/KO] IHCs. Scale bar = 10 μm. (C) Maximal projection of confocal sections from organs of Corti co-labeled for the presynaptic marker and anchor of the ribbon, bassoon (left column), and the postsynaptic marker, PSD-95 (middle column), in RBE[WT/WT] and RBE[KO/KO] IHCs. The merged picture (right column) shows the juxtaposition of bassoon (magenta) with PSD-95 (green), indicating its presence both at RBE[WT/WT] and ribbonless RBE[KO/KO] IHC synapses. Scale bar = 1 μm. (D) Maximal projection of confocal sections from organs of Corti co-labeled for the presynaptic marker RIM2 (left column) and the postsynaptic marker PSD-95 (middle column). The merged picture (right column) shows the co-localization of RIM2 (magenta) with PSD-95 (green) meaning its presence at the ribbonless IHC pre-synapses (scale bar = 5 μm). (E) Maximal projection of confocal sections from organs of Corti co-labeled for piccolino, a specific short splice variant of piccolo found at ribbons of RBE[WT/WT] IHC synapses (left column), co-labeled with PSD-95 (middle column). The merged pictures (right column) show PSD-95 (green) immunofluorescence lacking juxtaposed piccolino signal (magenta) in RBE[KO/KO] (bottom row), indicating absence of piccolino from afferent synapses of mutant IHCs. The punctate labeling for piccolo, away from PSD-95, most likely represents labeling of piccolo at conventional efferent synapses (schematically shown in *Figure 1A*). Scale bar = 5 μm. (F) Quantification of the area of PSD-95 immunofluorescent spots. The PSD-95 spots are siginifically bigger in the RBE[KO/KO] IHCs (p<0.0001, Mann-Whitney-Wilcoxon test, *n* = 178 spots, N = 3 for RBE[KO/KO] and *n* = 163 spots, N = 3 for RBE[WT/WT]). Box plots show 10, 25, 50, 75 and 90[th] percentiles with individual data points overlaid; means are shown as crosses.

DOI: https://doi.org/10.7554/eLife.29275.003

The following figure supplement is available for figure 1:

**Figure supplement 1.** Gene-dosage dependent expression of the RIBEYE at IHC AZs.

DOI: https://doi.org/10.7554/eLife.29275.004

dimensional Gaussian function to each PSD-95 immunofluorescent spot, revealing a significant increase in the RBE[KO/KO] condition (2.82 ± 0.09 μm², S.D. = 1.25; *n* = 178 spots, N = 3 vs. 1.74 ± 0.05 μm², S.D. = 0.58; *n* = 163 spots, N = 3 in RBE[WT/WT] IHCs; p<0.0001, Mann-Whitney-Wilcoxon; *Figure 1F*).

In order to study the effects of RIBEYE deletion on the ultrastructure of afferent IHC synapses, we performed transmission electron microscopy on random sections and electron tomography. Random ultrathin (70–75 nm) sections prepared from P21 mice (two animals per genotype) after aldehyde fixation and conventional embedding procedures showed that IHCs from RBE[KO/KO] mice completely lack synaptic ribbons, while RBE[WT/WT] and heterozygous RBE[WT/KO] typically display one ribbon per AZ (*Figure 2A–C*). Interestingly, ribbons of RBE[WT/KO] IHCs were smaller in height, width and area compared to RBE[WT/WT] IHC synaptic ribbons (*Figure 2—figure supplement 1A–C*; ribbon height: 118.32 ± 3.17 nm, S.D. = 31.84 nm; *n* = 101 ribbons, N = 2 for RBE[WT/KO] vs. 197.09 ± 4.36 nm, S.D. = 44.93 nm; *n* = 106 ribbons, N = 2 for RBE[WT/WT]; ribbon width: 119.80 ± 6.23 nm, S.D. = 62.27 nm for RBE[WT/KO] vs. 168.34 ± 6.83 nm, S.D. = 70.27 nm for RBE[WT/WT]; ribbon area: 11.5e3 ± 6.2e2 nm², S.D. = 6.3e3 nm² for RBE[WT/KO] vs. 25.4e3 ±1.1e2 nm², S.D. = 1.1e3 nm² for RBE[WT/WT]; p<0.0001, Mann-Whitney-Wilcoxon test for all) agreeing with the significantly reduced ribbon immunofluorescence intensity in the RBE[WT/KO] condition (see above and *Figure 1—figure supplement 1A–B*).

Random sections of synaptic contacts of RBE[KO/KO] mice (*Figure 2C*) often showed more than one presynaptic density (PD), each associated with a cluster of synaptic vesicles (henceforth considered individual AZs). The multiple AZs typically faced one continuous PSD, which is different from the synapses of immature IHC synapses that show multiple appositions of pre- and postsynaptic densities (*Sendin et al., 2007*; *Wong et al., 2014*). Moreover, we found more than one PD per synaptic contact in IHCs of older RBE[KO/KO] mice (*Figure 2E,F*; 6 weeks and 8 months, respectively), arguing against a delayed synaptic maturation to be the cause of the phenotype. Sections from tangential cuts of the synapse (*Figure 2D*), reconstructions from serial ultrathin sections (*Figure 2G,G'*) and quantifications of random sections (*Figure 2H*) corroborated the notion of multiple small ribbonless AZs at the synaptic contacts of RBE[KO/KO] IHCs. Analysis based on serial 3D reconstructions of synaptic contacts of RBE[KO/KO] IHCs from P21 animals showed on average 1.92 ± 0.34 PDs (S.D. = 1.16; *n* = 17 serial 3D reconstructions, N = 2) and 20.58 ± 2.98 total SVs per contact, S.D. = 10.34 (*Figure 2I*). The lateral extent of the individual PDs, determined in random sections, was comparable between RBE[KO/KO] and RBE[WT/WT] synapses (129.89 ± 2.53 nm, S.D. = 26.26 nm; *n* = 108 PDs, N = 2 for RBE[KO/KO] vs. 129.35 ± 4.89 nm, S.D. = 50.86 nm; *n* = 108 PDs, N = 2 for RBE[WT/WT]; p=0.92, NPMC test), while that of RBE[WT/KO] was enlarged (*Figure 2M*; 157.64 ± 7.19 nm, S.D. = 72.24 nm; *n* = 101 PDs, N = 2; p=0.0004 for comparison to RBE[WT/WT], NPMC test). PSDs tended to be increased in length at RBE[KO/KO] synapses compared to RBE[WT/WT] PSDs and were significantly larger

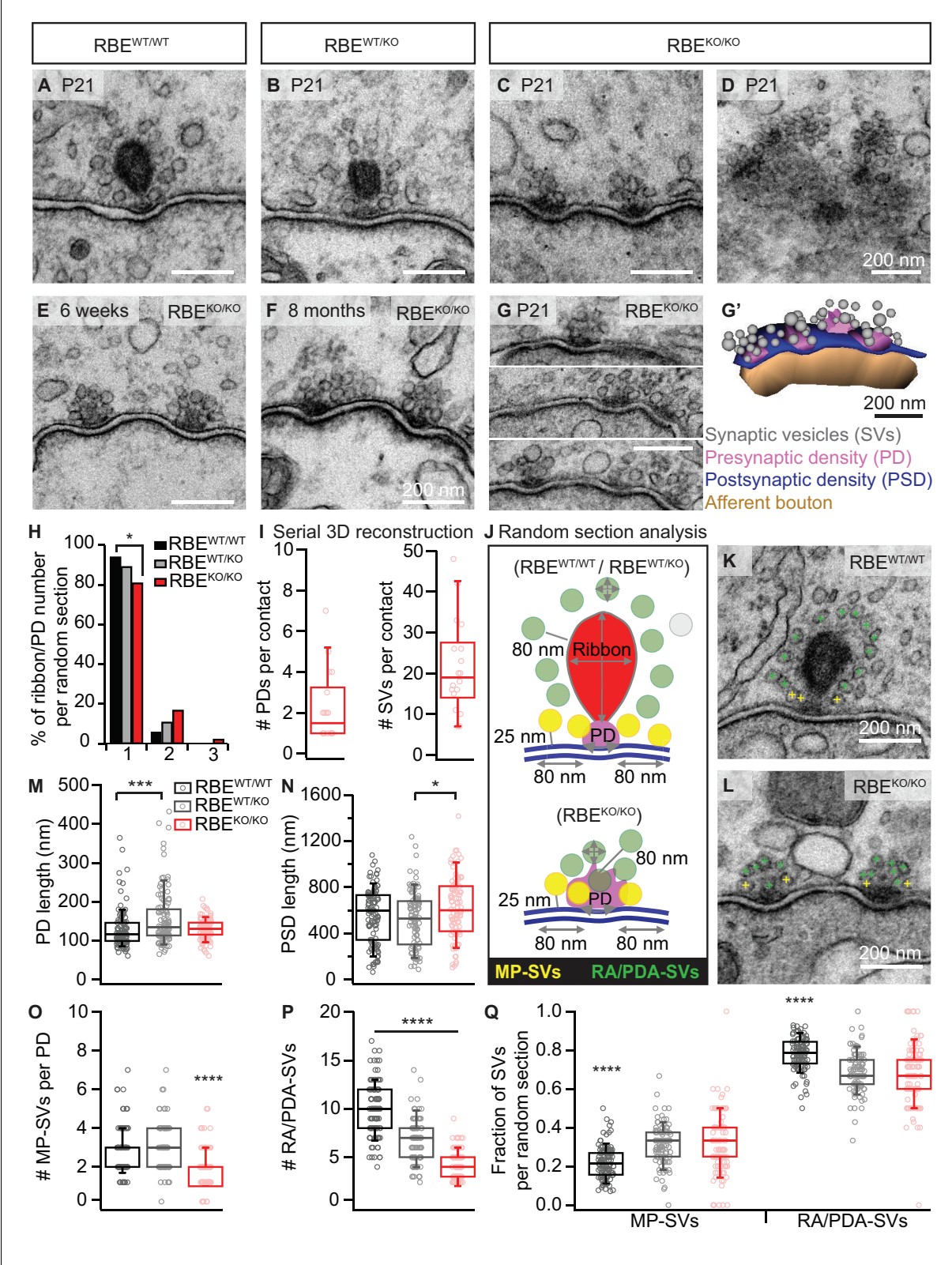

**Figure 2.** RIBEYE disruption transforms IHC synapses into contacts with multiple small ribbonless AZs. (A–C) Representative electron micrographs of IHC afferent synapses from P21 RBE$^{WT/WT}$, RBE$^{WT/KO}$ and RBE$^{KO/KO}$ mice. Ribbonless RBE$^{KO/KO}$ synapses display one or more presynaptic densities (PD) clustering SVs. Scale bars = 200 nm. (D) RBE$^{KO/KO}$ IHC AZ cut tangentially, revealing multiple PDs (here six) per AZ. (E, F) Representative electron micrographs of RBE$^{KO/KO}$ IHC synapses from mice at 6 weeks (E) and 8 months (F) of age: the presence of at least 2 AZs per contact in mature IHCs

*Figure 2 continued*

argues against a developmental delay. (G) Consecutive serial sections of a typical RBE$^{KO/KO}$ P21 IHC synapse showing multiple AZs. Scale bar = 200 nm. (G') Corresponding serial 3D reconstruction of the synapse in (G) showing four PDs (magenta) surrounded by a total of 48 SVs (gray). Scale bar = 200 nm. (H) Quantification of the number of ribbon/PD per random section. AZs with a single PD are less frequently observed in RBE$^{KO/KO}$ IHCs (*n* = 108 AZs, N = 2 for RBE$^{KO/KO}$ and *n* = 106 AZs, N = 2 for RBE$^{WT/WT}$; p<0.05, NPMC test). (I) Number of PDs and SVs per AZ in P21 RBE$^{KO/KO}$ mice in serial 3D reconstructions of RBE$^{KO/KO}$ afferent synapses. Box plots show 10, 25, 50, 75 and 90$^{th}$ percentiles with individual data points overlaid, as for (M, N, O, P & Q). (J) Schematic drawing illustrating the quantitative analysis of random sections. SV diameter: average of vertical and horizontal measurements from outer rim to outer rim. The ribbon height, width and area were measured as indicated by the gray lines. The length of the PD was determined along the AZ. For ribbon-occupied AZs: Membrane-proximal (MP) SVs (yellow) were counted in a distance of ≤25 nm from the AZ membrane (blue) and ≤80 nm from the PD. Ribbon-associated (RA) SVs were found in the first layer around the ribbon (red) with a maximum distance of 80 nm to the ribbon, quantified as indicated by the gray lines. For ribbonless AZs: Instead of RA-SVs we defined PD-associated SVs (PDA-SVs: all SV at PD with a maximum distance of 80 nm to the PD not matching the MP-SV criteria, defined as above). (K, L) Electron micrographs illustrating the quantification of the MP-SVs (yellow crosses) and the RA/PDA-SVs (green crosses). (M–Q) Quantification of random IHC synapse (P$_{21}$) sections revealed no significant differences between RBE$^{KO/KO}$ and RBE$^{WT/WT}$ for the PD and PSD length (PD length: *n* = 108 PDs, N = 2 for RBE$^{KO/KO}$ and *n* = 108 PDs, N = 2 for RBE$^{WT/WT}$; p=0.92, NPMC test and PSD length: *n* = 98 PSDs, N = 2 for RBE$^{KO/KO}$ and *n* = 113 PSDs, N = 2 for RBE$^{WT/WT}$; p=0.11, Tukey's test). However, in the RBE$^{WT/KO}$ IHCs, the PDs were bigger than in the WT IHCs (*n* = 101 PDs, N = 2; p=0.0004, NPMC test), and the PSDs were smaller than in the knock-out IHCs (*n* = 100 PSDs, N = 2 for RBE$^{WT/KO}$; p=0.01, Tukey's test). MP-SVs (*n* = 108 AZs, N = 2 for RBE$^{KO/KO}$, *n* = 106 AZs, N = 2 for RBE$^{WT/WT}$; p<0.0001, NPMC test) and RA/PDA-SVs (*n* = 108 AZs, N = 2 for RBE$^{KO/KO}$, *n* = 106 AZs, N = 2 for RBE$^{WT/WT}$; p<0.0001, Tukey's test) per AZ, as well as the fraction of RA/PDA-SVs in RBE$^{KO/KO}$, were significantly reduced (*n* = 108 AZs, N = 2 for RBE$^{KO/KO}$, *n* = 106 AZs, N = 2 for RBE$^{WT/WT}$; p<0.0001, NPMC test).

DOI: https://doi.org/10.7554/eLife.29275.005

The following figure supplement is available for figure 2:

**Figure supplement 1.** Random section analysis showed smaller synaptic ribbons and vesicles in RBE$^{WT/KO}$ mice.

DOI: https://doi.org/10.7554/eLife.29275.006

than RBE$^{WT/KO}$ PSDs (*Figure 2N*; 623.77 ± 26.70 nm, S.D. = 264.33 nm; *n* = 98 PSDs, N = 2 for RBE$^{KO/KO}$ vs. 555.91 ± 22.24 nm, S.D. = 236.42 nm; *n* = 113 PSDs, N = 2 for RBE$^{WT/WT}$ vs. 521.34 ± 24.20 nm, S.D. = 242.03 nm; *n* = 100 PSDs, N = 2 for RBE$^{WT/KO}$; p=0.01 for RBE$^{KO/KO}$ vs. RBE$^{WT/KO}$, Tukey's test), which is consistent with the greater area of PSD-95 immunofluorescent spots in the knock-out condition (*Figure 1E*).

In the following, we characterized the populations of presynaptic SVs in random sections of vertically-cut IHC synapses. We counted membrane-proximal SVs (MP-SVs, ≤25 nm distance between SV membrane and plasma membrane, laterally within 80 nm of the PD, yellow in *Figure 2J–L*) as well as ribbon-associated SVs (RA-SVs, first layer of SVs around the ribbon within 80 nm, green in *Figure 2J, K*) or 'PD-associated' SVs (PDA-SVs, ribbonless AZs: SVs within 80 nm distance of the PD and not falling into the MP-SV pool (see above), green in *Figure 2J,L*). We found both MP-SVs (*Figure 2O*; 1.92 ± 0.09, S.D. = 0.93; *n* = 108 AZs, N = 2 for RBE$^{KO/KO}$ vs. 2.99 ± 0.12, S.D. = 1.18; *n* = 101 AZs, N = 2 for RBE$^{WT/KO}$ vs. 2.77 ± 0.12, S.D. = 1.18; *n* = 106 AZs, N = 2 for RBE$^{WT/WT}$; p<0.0001 for RBE$^{KO/KO}$ vs. RBE$^{WT/WT}$, NPMC test) and PDA-SVs (*Figure 2P*; 4.12 ± 0.15, S.D. = 1.50; *n* = 108 AZs, N = 2 for RBE$^{KO/KO}$ vs. 10.09 ± 0.27, S.D. = 2.75; *n* = 106 AZs, N = 2 for RBE$^{WT/WT}$; p<0.0001, Tukey's test) of the individual ribbonless IHC AZs of RBE$^{KO/KO}$ mice to be significantly fewer than the corresponding number of MP-SVs and RA-SVs counted at RBE$^{WT/WT}$ AZs. The fraction of PDA-SVs relative to the total number of SVs at RBE$^{KO/KO}$ AZs was less than that of RA-SVs at RBE$^{WT/WT}$ AZs (*Figure 2Q*; 0.67 ± 0.02, S.D. = 0.16; *n* = 108 AZs, N = 2 for RBE$^{KO/KO}$ vs. 0.78 ± 0.01, S.D. = 0.08; *n* = 106 AZs, N = 2 for RBE$^{WT/WT}$; p<0.0001, NPMC test). Consequently, we observed an increase in the fraction of MP-SVs at RBE$^{KO/KO}$ AZs (*Figure 2Q*; 0.33 ± 0.02, S.D. = 0.16; *n* = 108 AZs, N = 2 for RBE$^{KO/KO}$ vs. 0.22 ± 0.01, S.D. = 0.08; *n* = 106 AZs, N = 2 for RBE$^{WT/WT}$; p<0.0001, NPMC test). In line with the decreased ribbon size of RBE$^{WT/KO}$ AZs, we found a reduced number of RA-SVs, indicating a hypomorphic phenotype upon the loss of one allele of the RIBEYE gene. The SV diameter was unchanged for all three genotypes when jointly considering SVs of all categories in random sections (*Figure 2—figure supplement 1D*; 39.59 ± 0.21 nm, S.D. = 5.37 nm; *n* = 108 AZs, N = 2 for RBE$^{KO/KO}$ vs. 40.53 ± 0.14 nm, S.D. = 4.44 nm; *n* = 101 AZs, N = 2 for RBE$^{WT/KO}$ vs. 41.80 ± 0.13 nm, S.D. = 4.79 nm; *n* = 106 AZs, N = 2 for RBE$^{WT/WT}$; p=0.30, NPMC test). However, we found a subtle but significant SV-diameter reduction in RBE$^{KO/KO}$ and RBE$^{WT/KO}$ for MP-SVs (*Figure 2—figure supplement 1E*; 39.29 ± 0.34 nm, S.D. = 4.82 nm; *n* = 108 AZs, N = 2 for RBE$^{KO/KO}$ vs. 41.79 ± 0.26 nm, S.D. = 4.53 nm; *n* = 106 AZs, N = 2 for RBE$^{WT/WT}$; p<0.0001, NPMC test and

40.29 ± 0.25 nm, S.D. = 4.40 nm; $n$ = 101 AZs, N = 2 for RBE$^{WT/KO}$ vs. RBE$^{KO/KO}$; p=0.03, NPMC test) and for RA-/PDA-SVs (*Figure 2—figure supplement 1F*; 39.72 ± 0.27 nm, S.D. = 5.61 nm; $n$ = 108 AZs, N = 2 for RBE$^{KO/KO}$ vs. 41.81 ± 0.15 nm, S.D. = 4.86 nm; $n$ = 106 AZs, N = 2 for RBE$^{WT/WT}$; p<0.0001, NPMC test and RBE$^{KO/KO}$ vs. 40.63 ± 0.17 nm, S.D. = 4.45 nm; $n$ = 101 AZs, N = 2 for RBE$^{WT/KO}$; p=0.003, NPMC test and RBE$^{WT/KO}$ vs. RBE$^{WT/WT}$; p=0.02, NPMC test).

Next, to capture the synapses in a near-to-native state and to evaluate vesicle tethering, we performed electron tomography on 250 nm thick sections that were prepared with high-pressure freezing and freeze-substitution (HPF/FS) of organs of Corti from P21 mice (*Figure 3*). Tomography confirmed the absence of synaptic ribbons and the presence of multiple AZs per contact, each with a clear PD (*Figure 3B,D,F*). However, we note that the 250 nm thick sections did typically not fully cover the synaptic contact, which leads to an underestimation for the total number of SVs particularly for the spatially extended RBE$^{KO/KO}$ synapses. The PDs appeared roundish in the RBE$^{KO/KO}$ with MP-SVs closely arranged around the PD as found at the more elongated RBE$^{WT/WT}$ AZs (*Figure 3C, D*). For the tomograms, we followed the definition of MP-SV pool according to the 2D-random sections (*Figure 2*), but in addition we measured the MP-SVs also in a maximum distance of 50 nm from the AZ membrane and ≤100 nm from the PD (*Figure 3—figure supplement 1* and *supplementary file 1*). This was motivated by the presence of long tethers connecting SV and AZ membrane and was previously introduced (*Jung et al., 2015a*). Further, we distinguished between tethered and non-tethered SVs (*Figure 3G*, *Figure 3—figure supplement 1A*). There, we focused our analysis on tethers to the ribbon/PD, plasma membrane and those interconnecting two adjacent SVs (*Figure 3H,I,M,N*). We found a significant reduction in the number of MP-SVs per AZ in RBE$^{KO/KO}$ IHCs (*Figure 3J*; RBE$^{KO/KO}$ = 6.30 ± 0.86, S.D. = 2.87 MP-SVs; $n$ = 11 AZs, N = 3 vs. RBE$^{WT/WT}$ = 8.70 ± 0.82, S.D. = 2.45 MP-SVs; $n$ = 9 AZs, N = 3; p=0.04, Mann-Whitney-Wilcoxon test; significant also by the second analysis method: *Figure 3—figure supplement 1B*, *supplementary file 1*), while the fraction of tethered MP-SVs (No. of tethered MP-SVs/No. of all MP-SVs) was not significantly altered (*Figure 3K*; RBE$^{KO/KO}$ = 0.75 ± 0.07, S.D. = 0.24; $n$ = 11 AZs, N = 3 vs. RBE$^{WT/WT}$ = 0.65 ± 0.06, S.D. = 0.18; $n$ = 9 AZs, N = 3; p=0.30, t-test; *Figure 3—figure supplement 1C*, *supplementary file 1*). The majority of the MP-SVs were tethered via a single tether in both RBE$^{WT/WT}$ and RBE$^{KO/KO}$ IHCs. The fraction of MP-SVs with multiple (≥2) tethers was significantly larger in RBE$^{KO/KO}$ IHCs (*Figure 3K*; single-tethered MP-SVs: RBE$^{KO/KO}$ = 0.55 ± 0.06, S.D. = 0.19; $n$ = 11 AZs, N = 3 vs. RBE$^{WT/WT}$ = 0.61 ± 0.06, S.D. = 0.17; $n$ = 9 AZs, N = 3; p=0.81; multiple-tethered MP-SVs: RBE$^{KO/KO}$ = 0.20 ± 0.05, S.D. = 0.15; $n$ = 11 AZs, N = 3 vs. RBE$^{WT/WT}$ = 0.04 ± 0.02, S.D. = 0.05; $n$ = 9 AZs, N = 3; p=0.01, Tukey's test; *Figure 3—figure supplement 1C*, *supplementary file 1*). Further, and in line with analysis of random sections, the number of PDA-SVs per RBE$^{KO/KO}$ AZ was smaller than that of RA-SVs at RBE$^{WT/WT}$ AZs (*Figure 3O*; RBE$^{KO/KO}$: 9.30 ± 1.13, S.D. = 3.74 PDA-SVs; $n$ = 11 AZs, N = 3 vs. RBE$^{WT/WT}$: 30.33 ± 3.00, S.D. = 9.01 RA-SVs; $n$ = 9 AZs, N = 3; p<0.0001, Mann-Whitney-Wilcoxon test; *Figure 3—figure supplement 1*, *supplementary file 1*). However, the fraction of PDA-SVs tethered to the PD was not different from that of RA-SVs tethered to the ribbon (*Figure 3P*; RBE$^{KO/KO}$: 0.80 ± 0.06, S.D. = 0.19 tethered PDA-SV fraction; $n$ = 11 AZs, N = 3 vs. RBE$^{WT/WT}$: 0.70 ± 0.06, S.D. = 0.17 tethered RA-SV fraction; $n$ = 9 AZs, N = 3; p=0.12, t-test; *Figure 3—figure supplement 1F*, *supplementary file 1*).

Finally, tomography indicated unchanged SV diameters at RBE$^{KO/KO}$ AZs (*Figure 3L,Q*; MP-SV diameter: 50.17 ± 0.90 nm, S.D. = 2.95 nm; $n$ = 11 PDs, N = 3 for RBE$^{KO/KO}$ vs. 47.81 ± 0.60 nm, S.D. = 1.70; $n$ = 9 ribbons, N = 3 for RBE$^{WT/WT}$; p=0.06, Mann-Whitney-Wilcoxon test, *Figure 3—figure supplement 1D*, *supplementary file 1*; RA/RA-SV diameter: 49.71 ± 0.83 nm, S.D. = 2.75; $n$ = 11 PDs, N = 3 for RBE$^{KO/KO}$ vs. 49.80 ± 0.78 nm, S.D. = 2.35; $n$ = 9 ribbons, N = 3 for RBE$^{WT/WT}$; p=0.71, Mann-Whitney-Wilcoxon test; *Figure 3—figure supplement 1G*, *supplementary file 1*). We presume that differences in the comparison of RBE$^{KO/KO}$ and RBE$^{WT/WT}$ between the random section and electron tomography analysis primarily reflects the larger number of AZ analyzed by the former approach.

We then used confocal and stimulated emission depletion (STED) super-resolution immunofluorescence microscopy in order to study the abundance and spatial organization of presynaptic Ca$_V$1.3 Ca$^{2+}$-channels (*Neef et al., 2018*), which contribute more than 90% of the voltage-gated Ca$^{2+}$-influx into IHCs (*Platzer et al., 2000*; *Brandt et al., 2003*; *Dou et al., 2004*). Organs of Corti from 3-week-old RBE$^{KO/KO}$ and RBE$^{WT/WT}$ mice were processed in parallel for immunohistochemistry and imaging. Ca$_V$1.3 Ca$^{2+}$-channels remained clustered at RBE$^{KO/KO}$ AZs and were identified as Ca$_V$1.3

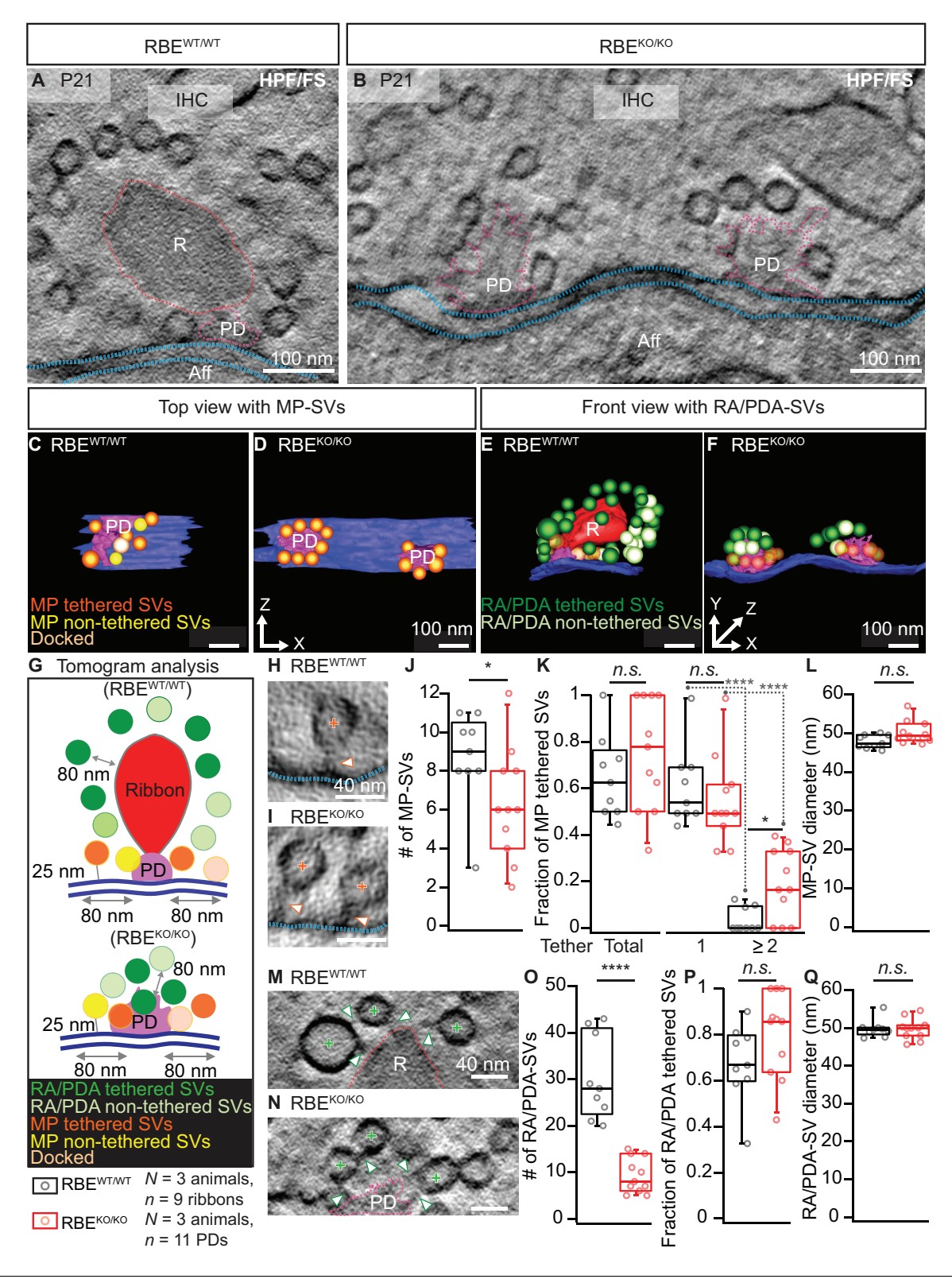

**Figure 3.** Electron tomography analysis of synaptic ultrastructure obtained after HPF/FS. (**A, B**) Exemplary virtual electron tomographic sections of P21 RBE^WT/WT (**A**) and RBE^KO/KO (**B**) highlight the ribbon R in red, the presynaptic density (PD) in magenta and the AZ membrane with blue dotted lines. No synaptic ribbons, but two PDs were observed in RBE^KO/KO (**B**). Scale bars = 100 nm. (**C–F**) 3D rendered models of RBE^WT/WT (**C, E**) and RBE^KO/KO (**D, F**) IHC synapses. (**C, D**) The top view depicts the MP-SV pool with tethered (orange), non-tethered (yellow) and docked (light orange) SVs. For clarity

*Figure 3 continued on next page*

*Figure 3 continued*

ribbons, RA/PDA-SVs are removed. Scale bars = 100 nm. (E,F) The front view shows the RA/PDA-SV pool from RBE$^{WT/WT}$ (E) and RBE$^{KO/KO}$ (F) IHCs. Tethered (dark green) and non-tethered (light green) RA/PDA-SVs. For the ease of visualization, the MP-SV pool is transparent here and other synaptic structures such as ribbon (red), PD (magenta) and AZ membrane (blue) are indicated. Scale bars = 100 nm. (G) Illustrations show the tomogram analysis parameters comparable to 2D-random section analysis (*Figure 2*), in addition to that the vesicle pools are subdivided into tethered, non-tethered and docked SVs. (H, I, M, N) Representative tomogram virtual sections of membrane-tethered MP-SVs (H, I; orange cross), ribbon/PD tethered SVs and ribbon/PD proximal interconnecting SVs (M, N; green cross) in RBE$^{WT/WT}$ (H, M) and in RBE$^{KO/KO}$ (I, N). Tethers are marked with a white arrowhead and other synaptic entities are color-coded similar to (A, B). Scale bars = 40 nm. (J–L) Quantification for the MP-SV pool is depicted; *n* = 9 ribbons, N = 3 animals for RBE$^{WT/WT}$ and *n* = 11 PDs, N = 3 animals for RBE$^{KO/KO}$. Fewer MP-SV were observed in RBE$^{KO/KO}$ (J; p=0.04, Mann-Whitney-Wilcoxon test). The fraction of tethered MP-SVs was unaltered in RBE$^{KO/KO}$ (K; p=0.30, t-test). Most of the SVs were tethered by a single tether in both RBE$^{KO/KO}$ and RBE$^{WT/WT}$. Significantly more SVs with multiple-tethers were observed in RBE$^{KO/KO}$ (K; single tethered MP-SVs, multiple-tethered MP-SVs: n.s.: p>0.05,*: p=0.01, ****: p<0.0001, Tukey's test). MP-SV diameter was unaltered in RBE$^{KO/KO}$ (L; p=0.06, Mann-Whitney-Wilcoxon test). (O–Q) Quantification for the RA/PDA-SVs, sample size is same as for the MP-SV analysis. Significantly fewer PDA-SVs were observed in RBE$^{KO/KO}$ (O; RBE$^{KO/KO}$: p<0.0001, Mann-Whitney-Wilcoxon test). The fraction of PDA tethered SVs in RBE$^{KO/KO}$ was comparable to RA tethered SVs in RBE$^{WT/WT}$ (P; p=0.12, t-test). SV diameters were unaltered in the RA/PDA vesicle pool (Q; p=0.06, Mann-Whitney-Wilcoxon test). Box plots show 10, 25, 50, 75 and 90$^{th}$ percentiles with individual data points overlaid. See *Figure 3—figure supplement 1* and *supplementary file 1* for modified tomogram analysis according to *Jung et al., 2015a*.

DOI: https://doi.org/10.7554/eLife.29275.007

The following figure supplement is available for figure 3:

**Figure supplement 1.** Electron tomogram analysis according to *Jung et al., 2015a*.
DOI: https://doi.org/10.7554/eLife.29275.008

labeling juxtaposed to PSD-95 immunofluorescent spots (*Figure 4A*). In order to analyze the spatial organization of synaptic Ca$^{2+}$-channels, we performed 3-color, 2D-STED immunofluorescence imaging for Ca$_V$1.3, bassoon (as a PD-marker), and PSD-95. While more than 80% of the RBE$^{WT/WT}$ synapses showed the typical stripe-like co-alignment of Ca$_V$1.3 and bassoon immunofluorescence (*Neef et al., 2018*), imaging of RBE$^{KO/KO}$ synapses indicated a high prevalence (over 70%) of smaller, rounder and often several Ca$^{2+}$-channel clusters and PDs per synaptic contact (*Figure 4B, C*). We then quantified stripe-like clusters by measuring their long and short axis using 2D Gaussian fits and found no differences between RBE$^{KO/KO}$ and RBE$^{WT/WT}$ AZs (*Figure 4D*). Finally, we quantified the number of Ca$_V$1.3-immunofluorescent structures per contact (as indicated by PSD-95 immunofluorescence). While more than 80% of RBE$^{WT/WT}$ synapses displayed a single cluster, over 60% of the RBE$^{KO/KO}$ synapses contained two or more Ca$^{2+}$-channel clusters (*Figure 4E*). Hence, the average number of Ca$_V$1.3-immunofluorescent structures was significantly higher at RBE$^{KO/KO}$ synapses compared to RBE$^{WT/WT}$ (2.06 ± 0.09, S.D. = 1.16; *n* = 178 spots, N = 3 vs. 1.16 ± 0.03, S.D. = 0.38; *n* = 183 spots, N = 2; p<0.0001, Mann-Whitney-Wilcoxon test) and we likely underestimated this difference due to the low resolution of 2D-STED in the z-axis. In summary, our results indicate that RIBEYE-disruption transforms the single ribbon-type AZ into a complex presynaptic organization with multiple conventional-like AZs facing the postsynaptic bouton.

## Altered operating range of presynaptic Ca$^{2+}$-influx at ribbonless IHC synapses

Next, we combined whole-cell patch-clamp with confocal Ca$^{2+}$-imaging of IHCs to study Ca$^{2+}$-influx at the whole IHC and single synapse levels using 5 mM [Ca$^{2+}$]$_e$ to augment the signal to noise. Using step-depolarizations in conditions that isolated the Ca$^{2+}$-current (see Materials and methods), we probed the amplitude and voltage-dependence of IHC Ca$^{2+}$-influx (*Figure 5A*). The amplitude of Ca$^{2+}$-influx (*Figure 5A$_i$*; for Ca$^{2+}$-current density, see *Figure 5A$_{ii}$*) was unaltered in RBE$^{KO/KO}$ IHCs (-151 ± 12.9 pA, S.D. = 59 pA; *n* = 21 IHCs, N = 8 in RBE$^{KO/KO}$ vs. -161 ± 15.4 pA, S.D. = 71 pA; *n* = 21 IHCs, N = 9 in RBE$^{WT/WT}$; p=0.62, t-test), in agreement with findings in retinal bipolar neurons (*Maxeiner et al., 2016*) but in contrast to our previous findings in ribbon-deficient IHCs of bassoon mutant mice (*Khimich et al., 2005*; *Frank et al., 2010*; *Jing et al., 2013*). Kinetics of Ca$^{2+}$-channel activation were unchanged (*Figure 5B*), whereas inactivation kinetics were slightly faster in the RBE$^{KO/KO}$ IHCs (smaller residual Ca$^{2+}$-current at 200 ms of depolarization (normalized to the peak current): 0.82 ± 0.007, S.D. = 0.02; *n* = 10 IHCs, N = 5 for RBE$^{KO/KO}$ vs. 0.85 ± 0.01, S.D. = 0.04; *n* = 11 IHCs, N = 7, in the RBE$^{WT/WT}$ condition; p=0.017, Mann-Whitney-Wilcoxon test; *Figure 5D*). When analyzing the voltage-dependence of Ca$^{2+}$-channel activation (*Figure 5C*), we found a small (2

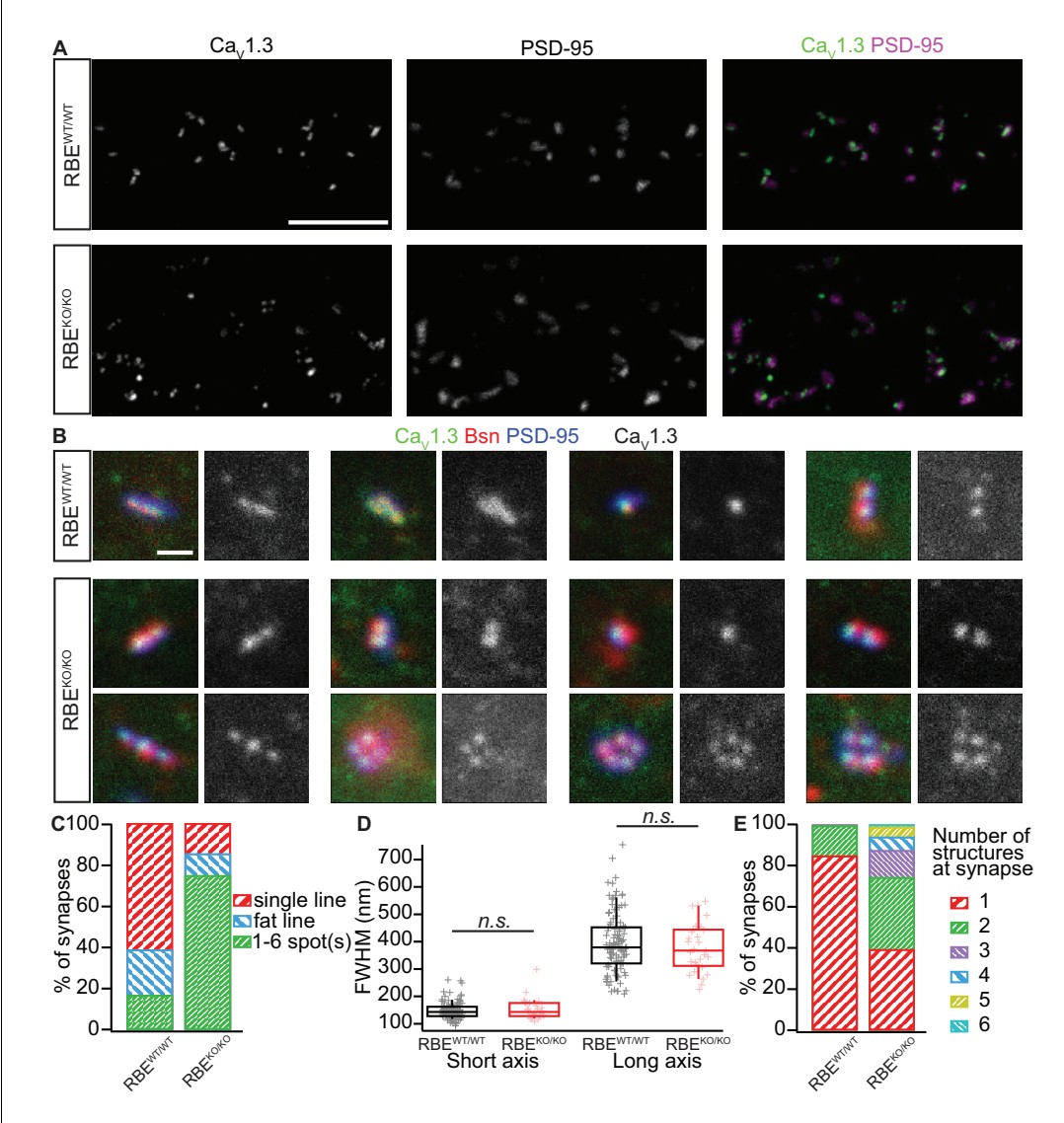

**Figure 4.** Analyzing AZ $Ca^{2+}$-channel clusters using confocal and STED immunofluorescence microscopy. (**A**) Maximal projections of confocal sections from organs of Corti immunolabeled for $Ca_V1.3$ $Ca^{2+}$-channels (left column) and PSD-95 (middle column). The merged pictures (right column) show their juxtaposition in both conditions. While we mostly found one linear/spot-like $Ca^{2+}$-channel cluster per PSD in $RBE^{WT/WT}$ IHCs (top row), we often observed several spots per PSD in $RBE^{KO/KO}$ IHCs (bottom row). Scale bar = 5 µm. (**B**) Triple co-labeling of $Ca_V1.3$ (green), bassoon (red), and PSD-95 (blue) at several IHC AZs from $RBE^{WT/WT}$ and $RBE^{KO/KO}$ mice, imaged in 2D-STED ($Ca_V1.3$ and PSD-95) and confocal mode (bassoon), showing that $Ca_V1.3$ $Ca^{2+}$-channels cluster at AZs in IHCs of both genotypes. $Ca_V1.3$ immunofluorescence is displayed in gray next to the merged image for better visualization. Scale bar = 500 nm. (**C**) 178 $RBE^{WT/WT}$ and 183 $RBE^{KO/KO}$ synapses were categorized according to the pattern of $Ca_V1.3$ immunofluorescence found by assigning them to a group of either line-shaped clusters, fat line-shaped clusters or one/multiple spots. A markedly higher fraction of synapses was found to display a spot-like $Ca_V1.3$-signal in $RBE^{KO/KO}$ than in $RBE^{WT/WT}$ IHCs. (**D**) Fitting of a 2D-Gaussian function to the immunofluorescence data of the line-shaped $Ca_V1.3$ clusters showed no difference in terms of size between $RBE^{WT/WT}$ and $RBE^{KO/KO}$ clusters, as estimated by the full width at half maximum of the Gaussian's short and long axis. Box plots show 10, 25, 50, 75 and $90^{th}$ percentiles with individual data points overlaid. (**E**) Quantification of the number of fluorescent structures (lines or spots) labeled by the anti-$Ca_V1.3$ antibody at $RBE^{WT/WT}$ and $RBE^{KO/KO}$ synapses showed a significantly increased number in the knockout (p<0.0001, Mann-Whitney-Wilcoxon test).
DOI: https://doi.org/10.7554/eLife.29275.009

mV) but significant depolarizing shift of the potential of half-maximal $Ca^{2+}$-channel activation, $V_h$ (*Figure 5C$_i$*, −22.96 ± 0.43 mV, S.D. = 2.39 mV; $n$ = 21 IHCs, N = 8 in $RBE^{KO/KO}$ vs. −25.04 ± 0.65 mV, S.D. = 2.98 mV; $n$ = 21 IHCs, N = 9 in $RBE^{WT/WT}$; p=0.017, t-test). When analyzed in a smaller data set recorded in 2 mM $[Ca^{2+}]_e$ the depolarized $V_h$-shift did not reach statistical significance (data

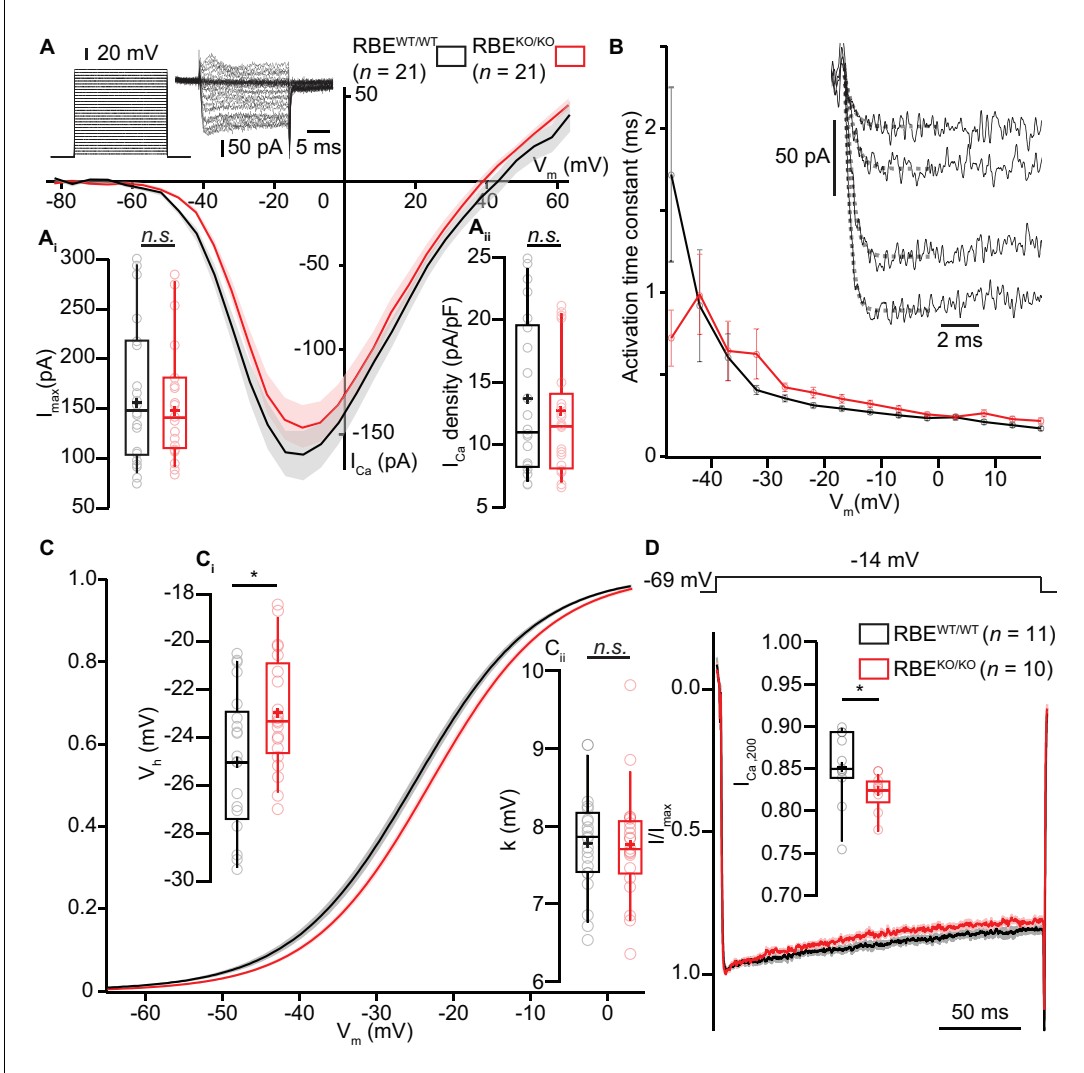

**Figure 5.** IHC $Ca^{2+}$-influx shows normal amplitude but a mild depolarized shift in voltage-dependence. (**A**) IV-relationship of the whole-cell $Ca^{2+}$-current in RBE[WT/WT] (black, $n = 21$ cells, N = 8) and RBE[KO/KO] (red, $n = 21$ cells, N = 8) IHCs show comparable (sign-inverted) current amplitudes (**A$_i$**, p=0.62, t-test) and density (**A$_{ii}$**, p=0.37, t-test.). Mean (line) ± S.E.M. (shaded areas) are displayed as for (**C, D**). The protocol, consisting of 20 ms steps of 5 mV from −82 to +63 mV, as well as exemplary resulting currents, aredisplayed in the left. Box plots show 10, 25, 50, 75 and 90[th] percentiles with individual data points overlaid, means are shown as crosses, as for (**C, D**). (**B**) Activation time constants (mean ± S.E.M.) of $Ca^{2+}$-currents at different potentials were obtained by fitting a power exponential equation to the first 5 ms of the current traces, revealing no differences between conditions. (**C**) Fractional activation of the whole-cell $Ca^{2+}$-current derived from the IV-relationships (**A**) was fitted to a Boltzmann function. (**Ci**) Box plots of the voltage for half-maximal activation $V_h$ and $V_h$-estimates of individual IHCs show a depolarized shift of the fractional activation of the $Ca_v1.3$ $Ca^{2+}$-channels in the RBE[KO/KO] IHCs (p=0.029, t-test). (**Cii**) Box plots of the voltage-sensitivity or slope factor k and k-estimates of individual IHCs illustrate comparable voltage sensitivity between both conditions (p=0.67, t-test). (**D**) Average peak-normalized $Ca^{2+}$-currents resulting from 200 ms depolarizations to −14 mV. We observe an enhanced inactivation in ribbonless IHCs, quantified as a reduced residual $Ca^{2+}$-current (inset). ($n = 10$ cells, N = 5 for RBE[KO/KO] and $n = 11$ cells, N = 7 in the RBE[WT/WT]; p=0.017, Mann-Whitney-Wilcoxon test).

DOI: https://doi.org/10.7554/eLife.29275.010

not shown). The average voltage-sensitivity of activation (slope factor k) was not altered (5 mM [$Ca^{2+}$]$_e$: p=0.67, t-test, **Figure 5C$_{ii}$**). Together, this suggests a RIBEYE/ribbons-mediated regulation of IHC $Ca^{2+}$-channels affecting their voltage-range of operation as well as their inactivation kinetics.

We then used the low-affinity $Ca^{2+}$-indicator dye Fluo-4FF (800 μM) to study $Ca^{2+}$-influx at individual IHC AZs (**Frank et al., 2009**) using a spinning-disk confocal microscope that allows rapid registering and recording of the majority of the IHC synapses (**Figure 6A**, **Ohn et al., 2016**). We chose conditions in which the $Ca^{2+}$-indicator fluorescence approximates synaptic $Ca^{2+}$-influx (**Frank et al.,**

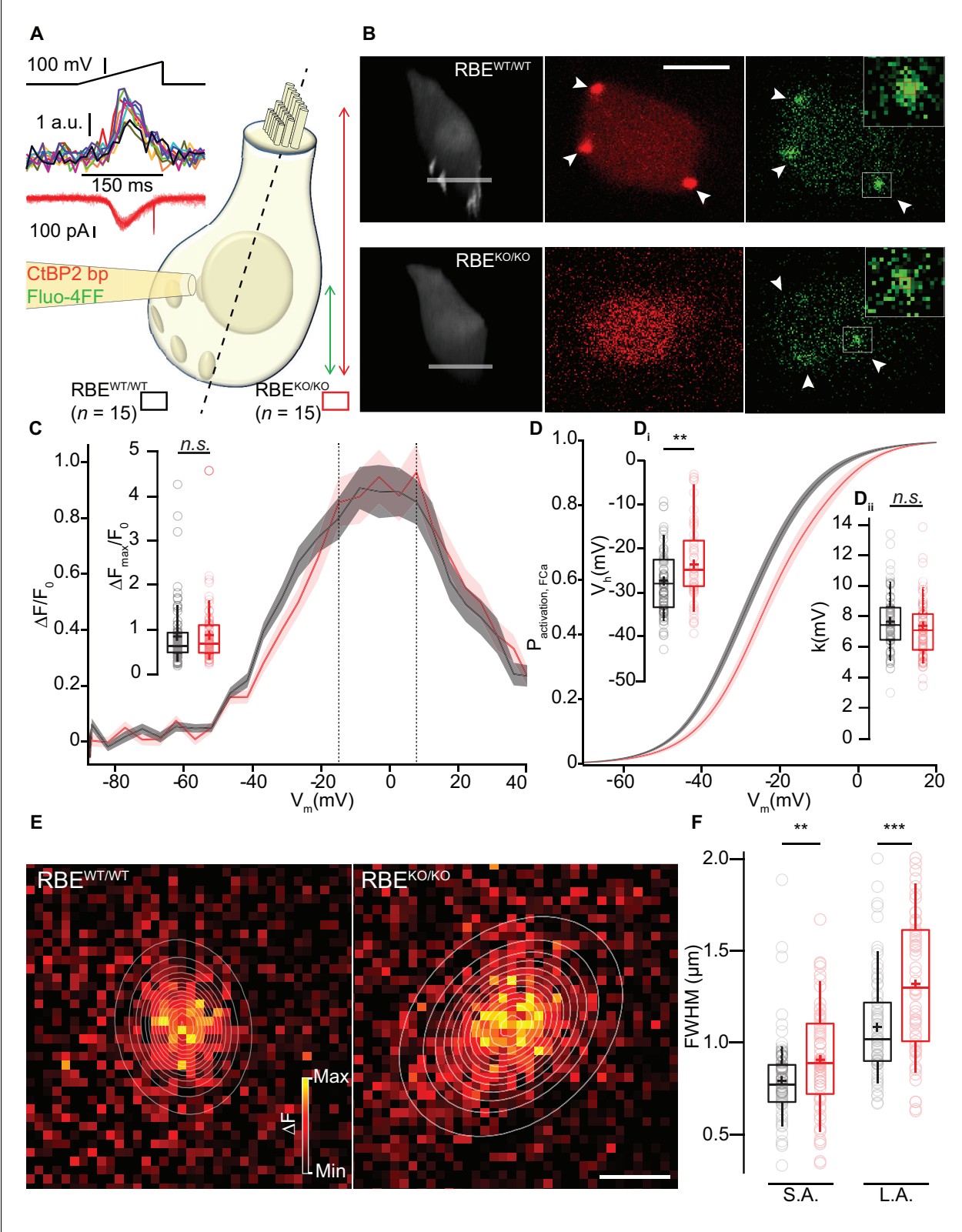

**Figure 6.** Synaptic Ca$^{2+}$-influx shows normal amplitude but shifted voltage-dependence and broader spread. (**A, B**) IHCs were patch-clamped at the modiolar basolateral face, loaded with TAMRA-CtBP2-binding peptide and the low affinity Ca$^{2+}$-indicator Fluo-4FF, and scanned in the red channel after loading for 4 min to image TAMRA-labeled ribbons, nuclei, and cytosol. 3D projection of TAMRA fluorescence shows the absence of ribbons in RBE$^{KO/KO}$ IHCs (B: 3D projection and red channel). Voltage-ramps from −87 to +63 mV during 150 ms (A: left top) were used to trigger

*Figure 6 continued on next page*

*Figure 6 continued*

synaptic hotspots of Fluo-4FF fluorescence (A: left middle, 10 AZs in one exemplary RBE$^{WT/WT}$ IHC, B: green channel, marked by arrowheads; $\Delta F$: average of the nine brightest pixels (red square)) and IHC Ca$^{2+}$-influx (A, left bottom). Ca$^{2+}$-imaging proceeded from the IHC bottom to the most apical ribbon in RBE$^{WT/WT}$, and from IHC bottom to +12 µm (typically reaching the bottom of nucleus) in RBE$^{KO/KO}$. Scale bar = 5 µm. (C) FV-relationship ($\Delta F/F_0$ vs. depolarization level in ramp, protocol as in A): approximating the voltage-dependence of synaptic Ca$^{2+}$-influx.Mean (line) ± S.E.M. (shaded areas) are displayed as for (D). (Ci) $\Delta F_{max}/F_0$ was calculated by averaging 5 values at the FV-peak (between the dotted lines) and was comparable between RBE$^{WT/WT}$ ($n$ = 78 AZs for 15 cells, N = 8) and RBE$^{KO/KO}$ IHCs ($n$ = 61 AZs for 15 cells, N = 7) (p=0.20, Mann-Whitney-Wilcoxon test). Box plots show 10, 25, 50, 75 and 90$^{th}$ percentiles with individual data points overlaid, means are shown as crosses, as for (D, F). (D) Fractional activation curves derived from fits to the FV-relationships (C) were fitted to a Boltzmann function. Mean (line) ± S.E.M. (shaded areas) are displayed. (Di) The voltage for half-maximal activation V$_h$ was significantly different between RBE$^{WT/WT}$ ($n$ = 68 AZs for 15 IHCs, N = 8) and RBE$^{KO/KO}$ ($n$ = 55 AZs for 15 IHCs, N = 7) AZs (p=0.0029, t-test), while the voltage-sensitivity or slope factor k (Dii) not (p=0.42, t-test). (E) Exemplary $\Delta F$ pictures of Fluo-4FF hotspots at RBE$^{WT/WT}$ (left) and RBE$^{KO/KO}$ (right) synapses fitted and overlaid by 2D-Gaussian functions to estimate spatial extent as full width at half maximum (FWHM) for the short axis (S.A.) and the long axis (L.A.). Scale bar = 1 µm. (F) Ribbonless synapses of RBE$^{KO/KO}$ IHCs showed a greater spatial spread of the Fluo-4FF fluorescence change. FWHM calculated from the Gaussian fitting to the Fluo-4FF fluorescence hotspot was larger for both axes in RBE$^{KO/KO}$ ($n$ = 61 AZs for 15 IHCs, N = 8) compared to RBE$^{WT/WT}$ ($n$ = 74 AZs for 15 IHCs, N = 7) (L.A.: p=0.00016; S.A.: p=0.0029, t-test).

DOI: https://doi.org/10.7554/eLife.29275.011

The following figure supplement is available for figure 6:

**Figure supplement 1.** Semi-quantitative immunofluorescence analysis of the three main proteinaceous Ca$^{2+}$-buffers.

DOI: https://doi.org/10.7554/eLife.29275.012

*2009*; *Ohn et al., 2016*) and henceforth refer to synaptic Ca$^{2+}$-influx when describing observations based on hotspots of Ca$^{2+}$-indicator fluorescence at the basolateral IHC membrane. Prior to analysis of synaptic Ca$^{2+}$-influx, we imaged fluorescently-conjugated CtBP2-binding peptide (*Zenisek et al., 2004*), which bound to the ribbon-occupied AZs in RBE$^{WT/WT}$ IHCs while it only caused nuclear and diffuse cytosolic fluorescence in the ribbonless RBE$^{KO/KO}$ IHCs (*Figure 6B*). We then employed ramp-depolarizations to assess amplitude and voltage-dependence of Ca$^{2+}$-influx at the synapses located in the subnuclear, basal part of the IHCs (*Figure 6A*). We found comparable maximal amplitudes of the baseline-normalized fluorescence change ($\Delta F/F_0$, 0.88 ± 0.08, S.D. = 0.66; $n$ = 61 AZs in 15 IHCs, N = 7 for RBE$^{KO/KO}$ vs. 0.85 ± 0.08, S.D. = 0.68; $n$ = 78 AZs in 15 IHCs, N = 8 for RBE$^{WT/WT}$; p=0.20, Mann-Whitney-Wilcoxon test; *Figure 6C*). This is compatible with an unaltered number of synaptic Ca$^{2+}$-channels at AZs of RBE$^{KO/KO}$ IHCs and consistent with our observations of normal whole-cell Ca$^{2+}$-current amplitudes. As previously reported (*Frank et al., 2009*; *Ohn et al., 2016*), there was a substantial variation of the maximal $\Delta F/F_0$ among the AZs, which was also comparable between AZs of both genotypes (c.v. = 0.75 for RBE$^{KO/KO}$ vs. c.v. = 0.80 for RBE$^{WT/WT}$).

Next, we analyzed the voltage-dependence of activation for the synaptic Ca$^{2+}$-influx as previously described (*Ohn et al., 2016*). Analysis of fractional activation revealed a depolarized shift in V$_h$ by on average 5 mV in RBE$^{KO/KO}$ IHCs ($-22.76 ± 1.25$ mV, S.D. = 9.26 mV; $n$ = 55 AZs in 15 IHCs, N = 7 for RBE$^{KO/KO}$ vs. $-27.37 ± 0.90$ mV, S.D. = 7.48 mV; $n$ = 68 AZs in 15 IHCs, N = 8 for RBE$^{WT/WT}$; p=0.0029, t-test; *Figure 6D,Di*), while the slope factor of voltage-dependent activation was unaltered (p=0.42, t-test, *Figure 6D$_{ii}$*). Such a shift in the operating range of synaptic Ca$^{2+}$-influx is expected to alter spontaneous and sound-evoked transmitter release (see below and *Ohn et al., 2016*). Finally, we studied the spatial extent of the synaptic Ca$^{2+}$-signals and estimated Full Width Half Maximum (FWHM) by fitting 2D Gaussian functions to the hotspots of Ca$^{2+}$-indicator fluorescence and found a greater spread of Ca$^{2+}$-signals at RBE$^{KO/KO}$ AZs (*Figure 6E,F*; long axis (L.A.) = 1317 ± 49 nm, S.D. = 384 nm, short axis (S.A.) = 906 ± 36 nm, S.D. = 284 nm; $n$ = 61 AZs in 15 IHCs, N = 7 vs. L.A. = 1083 ± 33 nm, S.D. = 283 nm; (p=0.00016, t-test), S.A. = 793 ± 27 nm, S.D. = 233 nm, (p=0.0029, t-test); $n$ = 74 AZs in 15 IHCs, N = 8 for RBE$^{WT/WT}$). This larger spread of the presynaptic Ca$^{2+}$-signals is in agreement with the presence of several Ca$_V$1.3-immunofluorescent clusters at RBE$^{KO/KO}$ synapses. In order to exclude lower IHC Ca$^{2+}$-buffering to contribute to the observed larger spread of presynaptic Ca$^{2+}$-signals, we performed semi-quantitative immunofluorescence analysis for the three major cytosolic Ca$^{2+}$-buffers, the EF-hand Ca$^{2+}$-binding proteins parvalbumin-$\alpha$, calretinin and calbindin-28k (*Pangršič et al., 2015*). We did not find any significant differences in their immunofluorescence intensity between IHCs of both genotypes (in arbitrary units, parvalbumin intensity: 2.24 ± 0.15, S.D. = 1.04 for RBE$^{KO/KO}$ vs. 1.88 ± 0.15, S.D = 1.01 for RBE$^{WT/WT}$, p=0.08; calbindin intensity: 0.82 ± 0.06, S.D. = 0.43 for RBE$^{KO/KO}$ vs.0.95 ± 0.07, S.D. = 0.49 for

RBE$^{WT/WT}$, p=0.23; calretinin intensity: 0.91 ± 0.04, S.D. = 0.26 for RBE$^{KO/KO}$ vs. 0.82 ± 0.04, S. D. = 0.28 for RBE$^{WT/WT}$, p=0.09; n = 49 cells and N = 4 for both conditions, Mann-Whitney-Wilcoxon test for all; *Figure 6—figure supplement 1*).

## IHC exocytosis is normal for strong depolarizations but mildly reduced for weak ones

The ribbon has been proposed to play a crucial role in the exocytosis of SVs at the IHC AZ (*Khimich et al., 2005*). Therefore, we monitored stimulated exocytosis of SVs with perforated-patch whole-cell recordings of exocytic membrane capacitance changes ($\Delta C_m$). Using IHCs from 2/3-week-old RBE$^{WT/WT}$ and RBE$^{KO/KO}$ mice, we found that $\Delta C_m$ in response to step-depolarizations to the potential that elicits maximal Ca$^{2+}$-influx (−14 mV) were not different between IHCs with or without ribbons. Both, fast exocytosis elicited by depolarizations of up to 20 ms, attributed to the fusion of the readily releasable pool of SVs (RRP, *Moser and Beutner, 2000*), and longer stimuli, thought to reflect sustained exocytosis, ongoing SV replenishment and fusion, were unaltered in RBE$^{KO/KO}$ IHCs (*Figure 7A,B,C*). On average, $\Delta C_m$ induced by 20 ms long maximal Ca$^{2+}$-influx was 16.70 ± 1.67 fF (S.D. = 5.80 fF; n = 12 cells, N = 7) for RBE$^{KO/KO}$ compared to 15.22 ± 0.98 fF (S.D. = 3.26 fF; n = 11 cells, N = 8) for RBE$^{WT/WT}$. Exocytic $\Delta C_m$ elicited by 200 ms long maximal Ca$^{2+}$-influx (same IHCs as for 20 ms), on average, amounted to 62.09 ± 5.40 fF (S.D. = 18.70 fF) for RBE$^{KO/KO}$ versus 63.28 ± 6.64 fF (S.D. = 22.04 fF) for RBE$^{WT/WT}$.

Moreover, trains of 20 step-depolarizations to −17 mV of 20 ms pulse duration did not reveal impaired exocytosis in RBE$^{KO/KO}$ IHCs, even when the inter-stimulus interval time was as short as 160 ms (*Figure 7D*; n = 11 cells, N = 5 for RBE$^{WT/WT}$ and n = 13 cells, N = 8 for RBE$^{KO/KO}$). We further explored RRP recovery from partial depletion using a paired-pulse protocol (two strong 20 ms depolarizations to −14 mV separated by 50, 110, 260 and 510 ms inter-pulse intervals; *Figure 7E,F*). RRP recovery, estimated as the $\Delta C_m$ ratio of the second and the first pulse, was not altered in RBE$^{KO/KO}$ IHCs at least when probing RRP exocytosis with maximal Ca$^{2+}$-influx from a hyperpolarized resting potential (*Figure 7F*). These data are in strong contrast to our previous findings in IHCs of bassoon mutant mice, which we had equivalently analyzed. There, the loss of synaptic ribbons, combined with a loss of functional bassoon resulted in profound deficits in exocytosis (*Khimich et al., 2005*; *Frank et al., 2010*; *Jing et al., 2013*).

Given the finding of a small depolarized shift in the operating range of Ca$^{2+}$-channels in RBE$^{KO/KO}$ IHCs (*Figure 6D*), we also probed the voltage-dependence of $\Delta C_m$ elicited by 100 ms step-depolarizations (*Figure 7G,H*). In agreement with the results obtained at maximal Ca$^{2+}$-influx, we did not find significant differences in $\Delta C_m$ for stronger depolarizations (e.g. pulses to −39 mV elicited an average $\Delta C_m$ of 20.67 ± 7.46 fF, S.D. = 23.58 fF, $n_{min}$ = 10 IHCs, N = 9 for RBE$^{KO/KO}$ vs. 24.12 ± 4.04 fF, S.D. = 13.98 fF, $n_{min}$ = 10 IHCs, N = 9 for RBE$^{WT/WT}$; p=0.20; Mann-Whitney-Wilcoxon test). However, for weaker depolarizations in the range of physiological receptor potentials (*Russell and Sellick, 1983*), we observed a subtle but significant reduction in exocytosis for RBE$^{KO/KO}$ IHCs (*Figure 7H*, p=0.0115, p=0.0295 and p=0.1321 for −45, –43 and −41 mV; without definitive outliers as determined by Graphpad Prism: p=0.0017, p=0.0042 and p=0.0489, respectively; Mann-Whitney-Wilcoxon test for all). For instance, depolarization to −45 mV elicited a $\Delta C_m$ of 4.79 ± 2.26 fF for RBE$^{KO/KO}$ (S.D. = 7.14 fF; $n_{min}$ = 10 cells, N = 9) compared to 9.85 ± 1.60 fF for RBE$^{WT/WT}$ (S. D. = 5.05 fF; $n_{min}$ = 10 cells, N = 8). The Ca$^{2+}$-current integral (Ca$^{2+}$-charge, $Q_{Ca}$), as well, tended to be reduced for RBE$^{KO/KO}$ IHC at these mild depolarizations, which, however, did not reach statistical significance (e.g. $Q_{Ca}$ for −45 mV: 3.90 ± 0.49 pC, S.D. = 1.54 pC for RBE$^{KO/KO}$ vs. 5.15 ± 0.54 pC, S.D. = 1.72 pC for RBE$^{WT/WT}$; p=0.1053; t-test). In summary, we found exocytosis to be unaltered for strong depolarizations but mildly decreased for more physiological stimuli in RBE$^{KO/KO}$ IHCs, which is in line with the findings of the companion paper by Becker *et al.*.

## Lack of synaptic ribbons impairs synchronous activation of the auditory pathway

Next, we studied sound encoding in RBE$^{KO/KO}$ mice *in vivo*. First, we recorded auditory brainstem responses (ABR) and found a significant reduction in the amplitude of wave I that reflects the SGN compound action potential (1.14 ± 0.13 µV, S.D. = 0.38 µV, N = 10 for RBE$^{KO/KO}$ vs. 3.30 ± 0.51 µV, S.D. = 1.54 µV, N = 10 for RBE$^{WT/WT}$, p=0.0007, NPMC test). This indicates less synchronous SGN

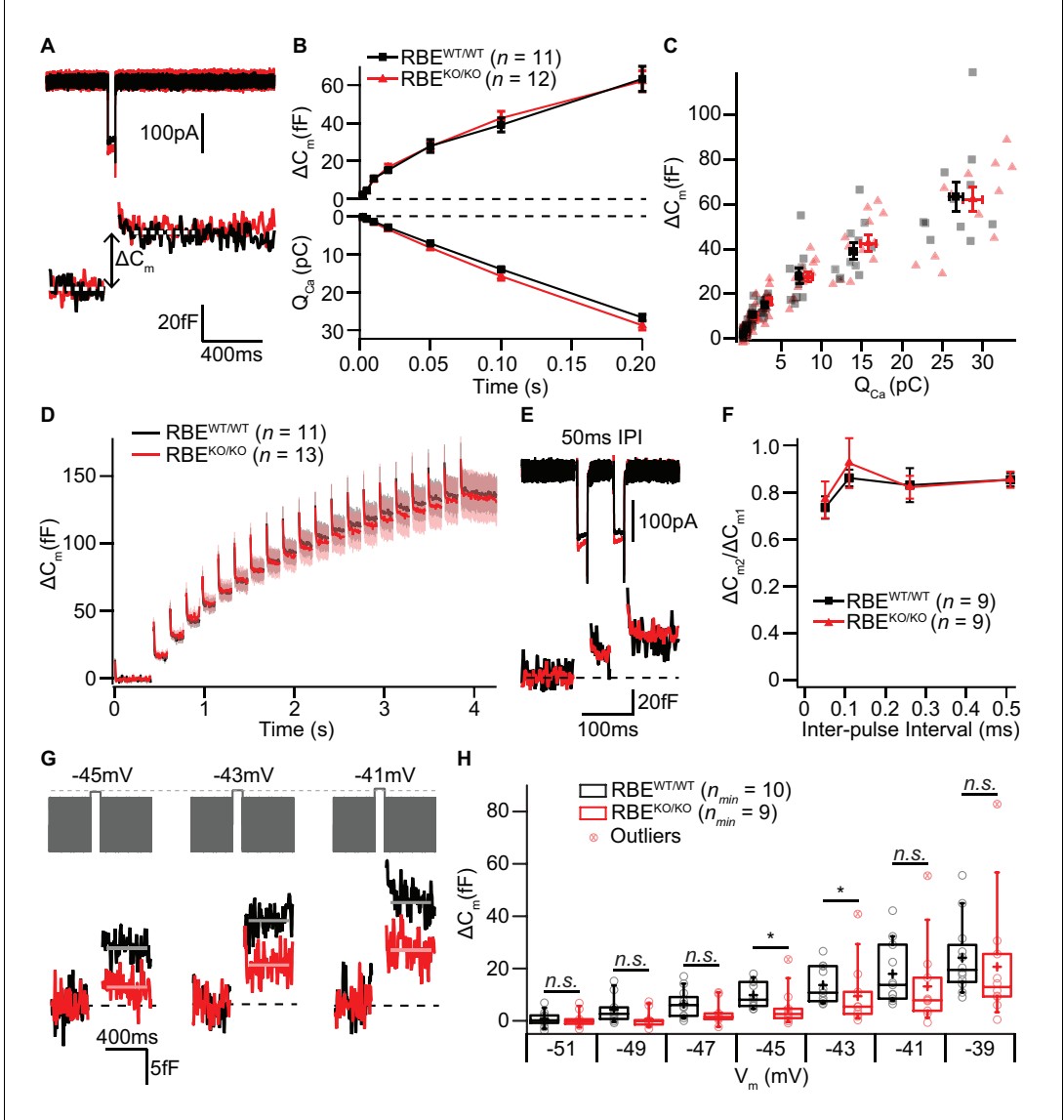

**Figure 7.** IHC exocytosis is normal for strong depolarizations but mildly reduced for weak ones. (A) Representative $Ca^{2+}$-currents (top) and corresponding low passed-filtered membrane capacitance ($\Delta C_m$) traces recorded from $RBE^{WT/WT}$ and $RBE^{KO/KO}$ IHCs upon 50 ms depolarizations from −69 to −14 mV. (B) Cumulative exocytosis (exocytic $\Delta C_m$, top) and corresponding $Ca^{2+}$-charge ($Q_{Ca}$, bottom) of $RBE^{WT/WT}$ ($n = 11$ cells, $N = 8$) and $RBE^{KO/KO}$ ($n = 12$ cells, $N = 7$) IHCs as a function of stimulus duration (2 to 200 ms to −14 mV) were unaltered in $RBE^{KO/KO}$ IHCs. Data is presented as mean ± S.E.M as for (F). (C) Relating $\Delta C_m$ to the corresponding $Q_{Ca}$ indicated comparable $Ca^{2+}$ efficiency of exocytosis between $RBE^{WT/WT}$ and $RBE^{KO/KO}$ IHCs. Mean ± S.E.M. for each pulse duration is presented in black and red; individual IHCs data points are overlaid. (D) Mean $\Delta C_m$ traces (shaded areas: S.E.M.) in response to trains of 20 ms depolarizations from −87 to −17 mV (20 stimuli separated by 160 ms) of $RBE^{WT/WT}$ ($n = 11$ cells, $N = 5$) and $RBE^{KO/KO}$ ($n = 13$ cells, $N = 8$) IHCs show comparable exocytic $\Delta C_m$. (E) Representative low pass-filtered $\Delta C_m$ traces in response to a pair of 20 ms pulses to −17 mV, separated by a 50 ms of inter-pulse interval (IPI). (F) Ratios of exocytosis ($\Delta C_{m2}/ \Delta C_{m1}$) to a pair of 20 ms pulses with varying inter-pulse intervals (50, 110, 260 and 510 ms) reveal a comparable recovery from RRP depletion between $RBE^{WT/WT}$ ($n = 9$ cells, $N = 6$) and $RBE^{KO/KO}$ IHCs ($n = 9$ cells, $N = 6$). (G) Representative low pass-filtered $\Delta C_m$ traces in response to 100 ms step-depolarizations to −45, −43 and −41 mV. (H) Box plot and single values of $\Delta C_m$ elicited by 100 ms step-depolarizations of $RBE^{WT/WT}$ ($n_{min} = 10$ cells, $N = 8$) and $RBE^{KO/KO}$ ($n_{min} = 9$ cells, $N = 9$) IHCs to different potentials. Exocytic $\Delta C_m$ of $RBE^{KO/KO}$ IHCs was reduced for mild depolarizations (−45, −43 and −41 mV; p=0.0115, p=0.0295 and p=0.1321, respectively; p=0.0017, p=0.0042 and p=0.0489, without definitive outliers; Mann-Whitney-Wilcoxon test), but comparable to $RBE^{WT/WT}$ IHCs at stronger depolarizations (−39 mV; p=0.2030, Mann-Whitney-Wilcoxon test). Box plots show 10, 25, 50, 75 and 90th percentiles with the individual data points overlaid.

DOI: https://doi.org/10.7554/eLife.29275.013

activation in the absence of synaptic ribbons (*Figure 8A,B*). The subsequent ABR waves (*Figure 8— figure supplement 1*) were normal in amplitude (waves II, IV and V, while wave III was reduced) indicating a degree of central auditory compensation for the sound encoding deficit, for example via coincidence detection of converging SGN input in the cochlear nucleus (*Joris et al., 1994*;

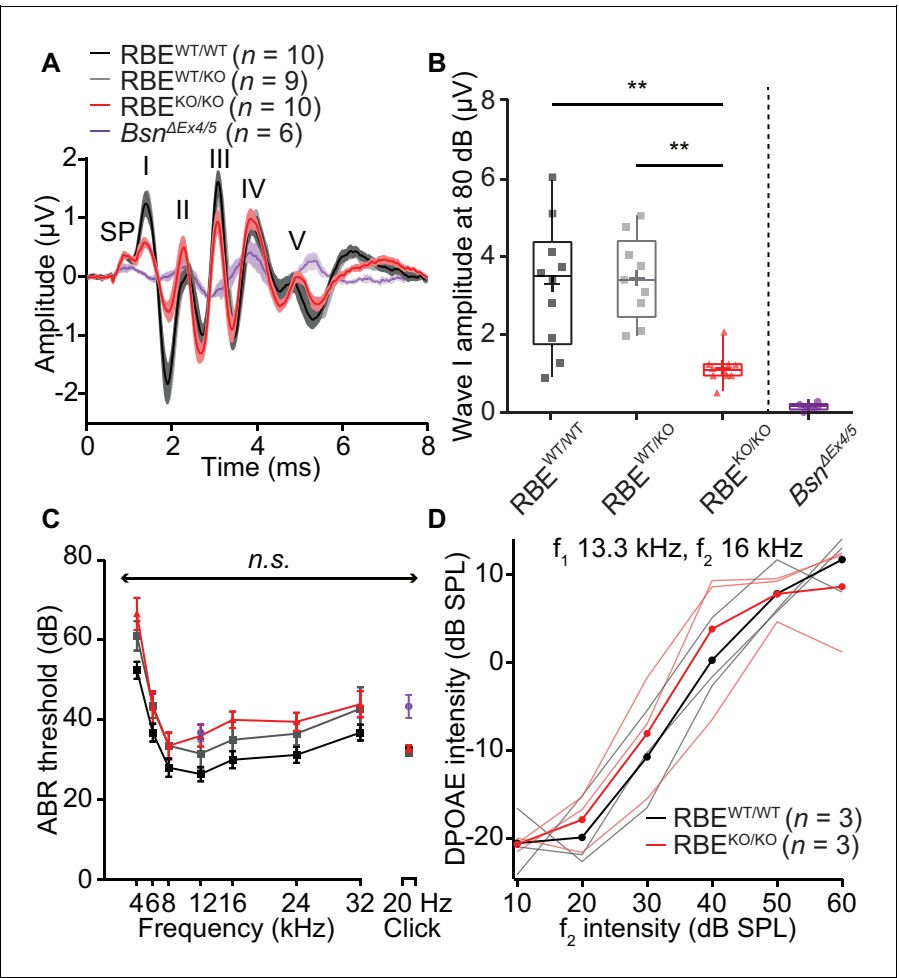

**Figure 8.** Auditory brainstem responses indicate impaired synchronous SGN activation in RBE[KO/KO] mice. (A) Average ABR waveforms in response to 80 dB clicks (*n* = N = 10 for RBE[KO/KO] and RBE[WT/WT], N = 9 for RBE[WT/KO] at 6 weeks of age) showed a reduced amplitude of ABR wave I in RBE[KO/KO] mice indicating an impairment of SGN activation, which is quantified in (B). This decrease was not as pronounced as in SGNs lacking bassoon (*Bsn[ΔEx4/5]*, data from *Jing et al., 2013*). The central ABR waves were better preserved, except for wave III. SP: summating potential (hair cell receptor potential), roman numerals (I–V): ABR waves generated along the early auditory pathway. Mean (lines) ± S.E.M. (shaded areas) are displayed. (B) ABR wave I was significantly reduced in RBE[KO/KO] mice as compared to RBE[WT/WT] and RBE[WT/KO] (p=0.0051 and p=0.0017, respectively, NPMC test). No statistical significance was observed between responses recorded in RBE[WT/WT] and RBE[WT/KO] mice (p>0.9999, NPMC test). Data from *Bsn[ΔEx4/5]* are shown for comparison. Box plots show 10, 25, 50, 75 and 90th percentiles with the individual data points overlaid, means are shown as crosses. (C) ABR thresholds were comparable in RBE[WT/WT], RBE[WT/KO] and RBE[KO/KO] for tone burst-driven (*n.s.*, Tukey's test) and click-driven ABRs (*n.s.*, NPMC test). Previously published data for *Bsn[ΔEx4/5]* showed elevated thresholds as a response to short stimuli (click 20 Hz) but a similar threshold level at 12 kHz. (D) DPOAE amplitude in response to pairs of simultaneous sine waves ($f_1$ and $f_2$, frequencies indicated on panel) at increasing stimulus intensity ($f_1$ intensity 10 dB above $f_2$ in all cases). Mean (thick lines) and data from individual mice (*n* = N = 3 in RBE[WT/WT] and RBE[KO/KO]) are displayed.

DOI: https://doi.org/10.7554/eLife.29275.014

The following figure supplement is available for figure 8:

**Figure supplement 1.** Quantification of ABR waves II-V amplitude in RBE[KO/KO] mice.

DOI: https://doi.org/10.7554/eLife.29275.015

*Strenzke et al., 2009*). We found a non-significant trend of ABR threshold to be increased across all frequencies in RBE$^{KO/KO}$ mice (approximately 10 dB across all frequencies, *Figure 8C*; refer to the companion paper Becker *et al.* showing significantly increased ABR-thresholds based on a larger sample, N = 28 RBE$^{KO/KO}$ mice vs. 22 RBE$^{WT/WT}$ mice). Cochlear amplification, probed by recordings of distortion product otoacoustic emissions (DPOAE, *Figure 8D*), was intact in RBE$^{KO/KO}$ mice. Additionally, RBE$^{WT/KO}$ mice showed no significant changes in ABR wave I amplitudes and ABR thresholds (*Figure 8*), suggesting that the subtle morphological differences observed for afferent synapses of RBE$^{WT/KO}$ IHCs by electron and confocal-immunofluorescence microscopy did not turn into a deficit of sound coding measurable by ABR recordings. The wave I amplitude reduction and ABR threshold elevation were much less pronounced than in bassoon mutant mice (*Khimich et al., 2005*; *Buran et al., 2010*; *Jing et al., 2013*).

We then turned to *in vivo* extracellular recordings from single auditory neurons by targeting glass microelectrodes to where the auditory nerve enters the anteroventral cochlear nucleus (AVCN) in the brainstem (*Taberner and Liberman, 2005*; *Jing et al., 2013*). 'Putative' SGNs (hereafter dubbed SGN for simplicity) were identified based on the depth of electrode position and their firing response to pure-tone stimulation (primary-like peristimulus time histogram and latency, *Figure 9*) and analyzed in separation from 'putative' cochlear nucleus neurons (*Figure 10*). Since all firing of the individual SGN is thought to be driven by transmitter release from a single IHC AZ (*Heil et al., 2007*; *Liberman, 1978*; *Robertson and Paki, 2002*), these recordings provide insight into single AZ function. We first assessed the spontaneous firing activity and found an increased abundance of SGNs with low spontaneous firing rates in RBE$^{KO/KO}$ mice (72% with rates < 10 Hz, *n* = 43 SGNs, N = 9 vs. 50% in RBE$^{WT/WT}$, *n* = 40 SGNs, N = 8; p=0.0267, Kolmogorov-Smirnov test; *Figure 9A*). Frequency tuning was intact in RBE$^{KO/KO}$ SGNs (*Figure 9B*): the sharpness of tuning expressed by the $Q_{10dB}$ (width of tuning curve 10 dB above threshold at the characteristic frequency ($C_f$) normalized by $C_f$) was comparable (mean: 9.28 ± 1.01, S.D. = 6.32 and median: 7.41 for RBE$^{KO/KO}$ SGNs, *n* = 39 SGNs, N = 9 vs. mean: 12.50 ± 1.98, S.D. = 11.91 and median: 8.36 for RBE$^{WT/WT}$ SGNs, *n* = 36 SGNs, N = 9; p=0.28, Mann-Whitney-Wilcoxon test). However, the sound threshold at $C_f$ was significantly elevated by almost 20 dB in RBE$^{KO/KO}$ mice (35.60 ± 3.45 dB SPL, S.D. = 22.66 dB SPL for RBE$^{KO/KO}$ SGNs, *n* = 43 SGNs, N = 9 vs. 16.05 ± 2.47 dB SPL, S.D. = 15.42 dB SPL for RBE$^{WT/WT}$ SGNs, *n* = 39 SGNs, N = 9, p<0.0001, Mann-Whitney-Wilcoxon test; *Figure 9C*). Given the normal frequency tuning and DPOAE, this threshold increase seems unlikely to result from a putative functional cochlear deficit upstream of the IHCs.

Next, we studied the firing response of SGNs to 50 ms tone bursts (at $C_f$ and 30 dB above sound threshold, 200 ms inter-stimulus interval), which is governed by the presynaptic glutamate release and postsynaptic spike generation. The peak firing rate at sound onset is thought to reflect the initial rate of release from the SV-occupied release sites of the RRP ('standing RRP', [*Oesch and Diamond, 2011*; *Pangršič et al., 2012*]). Refractoriness and the decline of release rate due to partial depletion of the standing RRP likely dominate the subsequent spike rate adaptation. Finally, the adapted firing rate reports SV replenishment and subsequent fusion (reviewed in *Pangršič et al., 2012*; *Rutherford and Moser, 2016*). We observed reduced spike rates of SGNs from RBE$^{KO/KO}$ mice (*Figure 9D,E*) both at sound onset (p=0.0001, *n* = 39 SGNs, N = 8 in RBE$^{KO/KO}$ and *n* = 38 SGNs, N = 9 in RBE$^{WT/WT}$, t-test) and after short-term adaptation (p=0.0023, Mann-Whitney-Wilcoxon test). Both, peak and adapted rates were similarly affected by the RIBEYE-disruption, indicated by the scatter plot of peak vs. adapted rates (*Figure 9E*). A significant peak rate reduction was also observed at higher stimulation frequencies (10 Hz, *Figure 9G–H*). The spike rates were better preserved in RBE$^{KO/KO}$ SGNs than in SGNs of bassoon mutant mice ($Bsn^{\Delta ex4/5}$ data of *Jing et al. (2013)*, purple data in *Figure 9G–I*). We approximated adaptation within the 50 ms response by single-exponential fitting since double exponential fitting did not regularly report two temporally discernible components in RBE$^{KO/KO}$ SGNs. The mean apparent adaptation time constant reported by single-exponential fitting were significantly slowed in RBE$^{KO/KO}$ SGNs (9.83 ± 0.50 ms, S.D. = 2.85 ms, median: 10.46 ms, *n* = 32 SGNs, N = 8) as compared to RBE$^{WT/WT}$ SGNs (8.71 ± 0.50 ms, S.D. = 3.05 ms, median: 8.73 ms, *n* = 37 SGNs, N = 9, p=0.033, Mann-Whitney-Wilcoxon test). The results of double-exponential fitting of RBE$^{WT/WT}$ and RBE$^{KO/KO}$ SGNs support the slowed adaptation kinetics and are presented in *Table 1*. As expected for the reduced peak firing rate, we found prolonged first spike latency which also showed greater temporal jitter (*Figure 9F*). The reduced peak firing rate together with increased first spike latency jitter likely explain the reduction in ABR

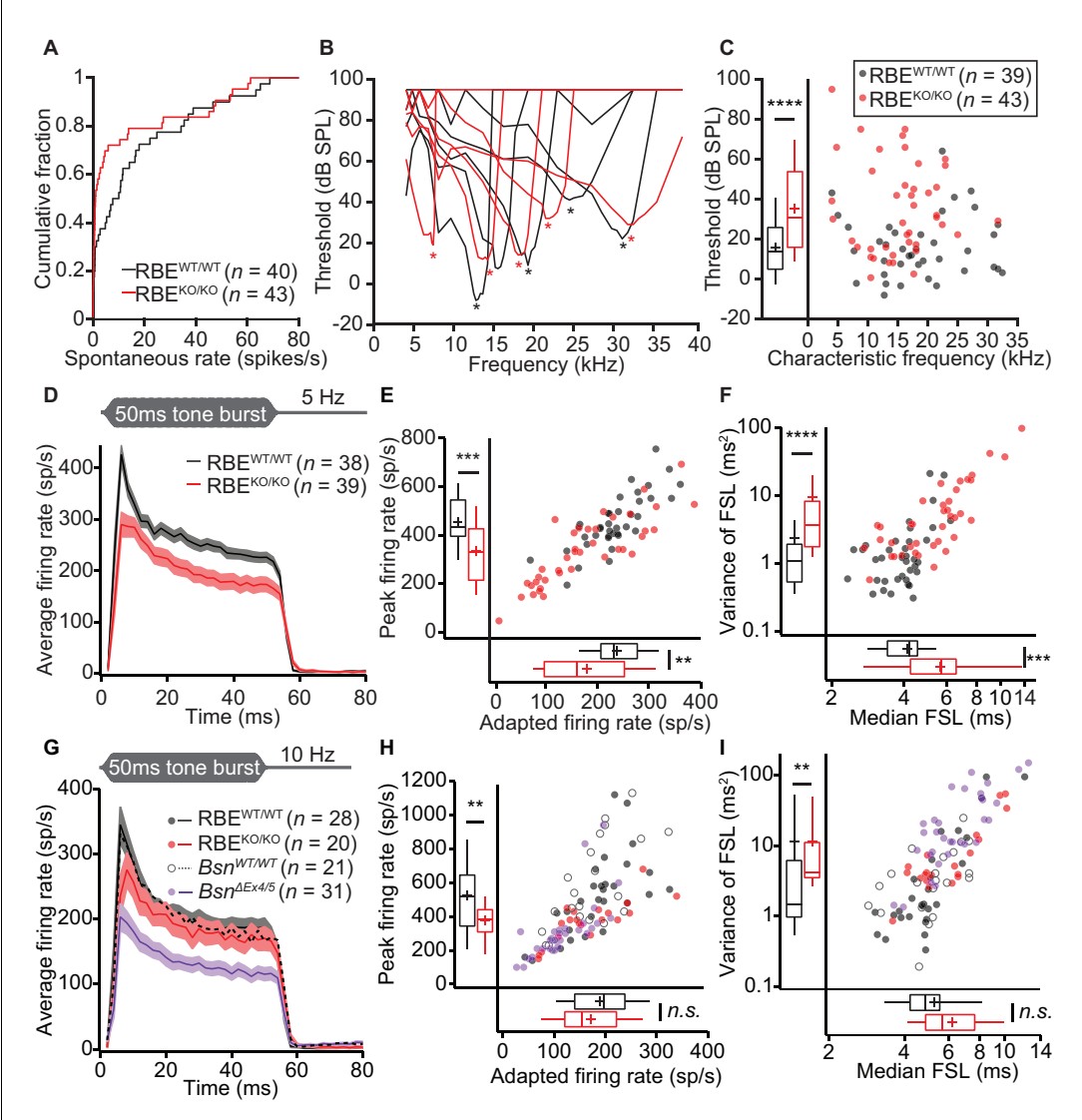

**Figure 9.** Impaired spontaneous and sound-evoked firing in putative SGNs of RBE[KO/KO] mice. (A) Cumulative distribution functions of spontaneous firing rates of putative SGNs showed a higher fraction of low spontaneous rate SGNs in RBE[KO/KO] mice ($n$ = 43 SGNs) than in RBE[WT/WT] mice ($n$ = 40 SGNs, N = 8) (p=0.027, Kolmogorov-Smirnov test). (B) Representative frequency tuning curves of RBE[KO/KO] and RBE[WT/WT] SGNs showed comparable sharpness of tuning between SGNs of both genotypes. * point to the characteristic frequency ($C_f$) for which an increase in SGN firing requires the least sound pressure level. (C) Thresholds at $C_f$ of RBE[KO/KO] SGNs ($n$ = 43 SGNs, N = 9) were higher than those in RBE[WT/WT] mice ($n$ = 39 SGNs, N = 9) (p<0.0001, Mann-Whitney-Wilcoxon test). Box plots show 10, 25, 50, 75 and 90[th] percentiles, means are shown as crosses, as for (E), (F), (H) and (I). (D) Average PSTH (bin width = 2 ms) of RBE[KO/KO] ($n$ = 39 SGNs, N = 8) and RBE[WT/WT] SGNs ($n$ = 38 SGNs, N = 9) recorded in response to 50 ms tone bursts at $C_f$ 30 dB above threshold at a stimulus rate of 5 Hz. The time course of adaptation of RBE[KO/KO] SGNs (calculated by fitting a single exponential function to the individual histograms) was significantly longer (p=0.033, Mann-Whitney-Wilcoxon test). Mean (line) ± S.E.M. (shaded areas) are displayed as for (G). (E) Scatterplot of peak firing rate (bin with highest rate at sound onset) and adapted firing rate (averaged 35–45 ms from response onset) revealed lower firing rates in RBE[KO/KO] ($n$ = 39 SGNs) as compared to RBE[WT/WT] ($n$ = 38 SGNs, N = 9) mice (data from (D), peak rate: p=0.0001, adapted rate: p=0.0023, Mann-Whitney-Wilcoxon test). (F) Increased latency (data from (D), p=0.0002) and variance of latency (p<0.0001, Mann-Whitney-Wilcoxon test) of the first spike after sound onset in RBE[KO/KO] SGNs indicated lower temporal precision of sound onset coding. (G) Average PSTH (bin width = 2 ms) of RBE[KO/KO] ($n$ = 20 SGNs, N = 6) and RBE[WT/WT] SGNs ($n$ = 28 SGNs, N = 8) were recorded in response to 50 ms tone bursts at $C_f$ 30 dB above threshold at a stimulus rate of 10 Hz and showed a similar adapted response in RBE[KO/KO] as compared to lower stimulation rates (*Figures 9,11*) but still a lower onset response than in RBE[WT/WT] SGNs. Responses in $Bsn^{\Delta Ex4/5}$ mutants (shown for comparison, re-plotted from *Jing et al., 2013*) were considerably lower. (H) Scatterplot of peak firing rate (bin with highest rate at sound onset) and adapted firing rate (averaged 35–45 ms from response onset) show decreased onset firing rates in RBE[KO/KO] as compared to RBE[WT/WT] mice (data from (G), p=0.0093, Mann-Whitney-Wilcoxon test). The adapted response was comparable in both cases (p=0.3584, t-test). Data points from $Bsn^{\Delta Ex4/5}$ mutants (re-plotted from *Jing et al., 2013*) and WT littermates are shown for comparison, also in (I). (I) Increased variance of first spike latency after sound onset in RBE[KO/

*Figure 9 continued on next page*

*Figure 9 continued*

KO SGNs (p=0.0089, Mann-Whitney-Wilcoxon test) and comparable latencies were observed at this stimulation frequency (data from (**G**), p=0.0761, Mann-Whitney-Wilcoxon test).

DOI: https://doi.org/10.7554/eLife.29275.016

wave I amplitude. The firing of putative AVCN neurons was better preserved: putative bushy cells showed normal sound driven rates and chopper cells only a mild reduction in peak rate (*Figure 10*).

Next, we explored the encoding of sound intensity by estimating the mean firing rate during 50 ms tone bursts at different sound pressure levels. These 'rate-level functions' (*Figure 11A*) indicated that the spike rate increase with the sound pressure level (p=0.068, $n$ = 24 SGNs, N = 8 in RBE$^{KO/KO}$ and $n$ = 19 SGNs, N = 7 in RBE$^{WT/WT}$, Mann-Whitney-Wilcoxon test, *Figure 11—figure supplement 1A*) and the dynamic range of sound coding (sound pressure level for which the spike rate changes from 10–90%, *Figure 11—figure supplement 1B*, p=0.3044, t-test) were not significantly altered. We then used transposed tones (C$_f$ at 500 Hz modulation frequency) in order to probe for the temporal fidelity and reliability of firing in RBE$^{KO/KO}$ SGNs in the steady state (*Figure 11D*). These experiments corroborated the reduced maximal firing rate of RBE$^{KO/KO}$ SGNs ($n$ = 22 SGNs, N = 7 in RBE$^{KO/KO}$ and $n$ = 15 SGNs, N = 6 in RBE$^{WT/WT}$, p<0.0001, t-test) and indicated that the temporal precision of sound coding is impaired also in the steady state (reduced Synchronization Index: p=0.0043, t-test).

In order to further scrutinize the potential role of the synaptic ribbon in vesicle replenishment, we studied the response to prolonged tone-stimulation (*Figure 11B*, 500 ms at C$_f$ and 30 dB above threshold, 2 s inter-stimulus interval). The peak rate was better preserved in RBE$^{KO/KO}$ SGNs than

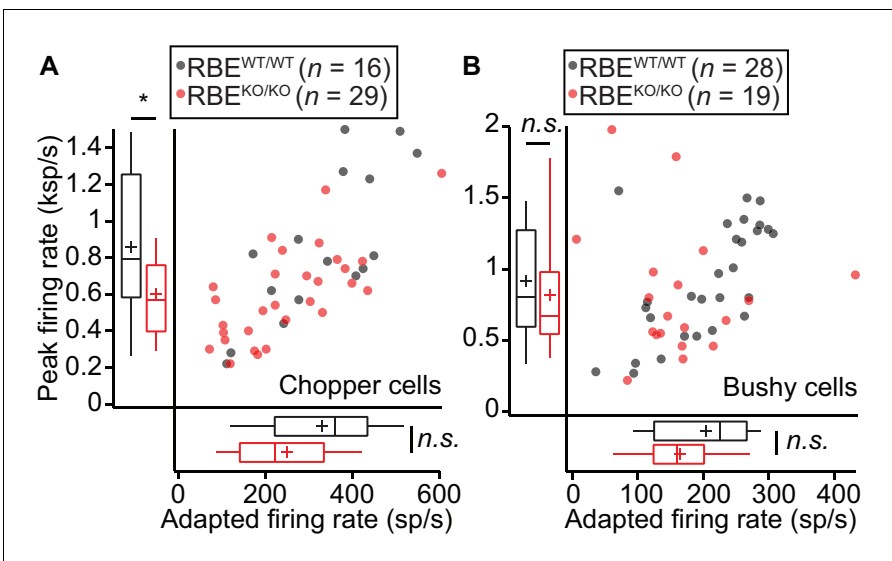

**Figure 10.** Responses to pure tones in cochlear nucleus cells showed that the mutant phenotype is partially compensated in higher stations of the auditory pathway. (**A**) Quantification of peak and adapted responses to 50 ms tone burst stimulation, 30 dB above threshold at C$_f$ in SGNs displaying a chopper discharge pattern (periodically alternating phases of high firing rates with low firing rates, typical of multipolar cells in the posterior ventral cochlear nucleus), showed that the differences in rate faded away opposite to those shown at the level of the auditory nerve. Peak rates were still significantly lower in RBE$^{KO/KO}$ ($n$ = 29 SGNs, N = 9) as compared to RBE$^{WT/WT}$ ($n$ = 16 SGNs, N = 9) mice (p=0.0303, Mann-Whitney-Wilcoxon test), while the adapted rate showed a non-significant trend towards reduction in RBE$^{KO/KO}$ (p=0.0538, t-test). Box plots show 10, 25, 50, 75 and 90$^{th}$ percentiles and means are shown as crosses, as for (**B**). (**B**) Same recordings paradigm as (**A**) performed in a fiber with bushy cell discharge pattern (similar to the one found in SGNs, typical also in this type of cochlear nucleus neurons) showed comparable responses in both peak (p=0.2601, Mann-Whitney-Wilcoxon test) and adapted rate (p=0.0510, Mann-Whitney-Wilcoxon test) in RBE$^{KO/KO}$ ($n$ = 19 SGNs, N = 7) and RBE$^{WT/WT}$ ($n$ = 28 SGNs, N = 10).

DOI: https://doi.org/10.7554/eLife.29275.017

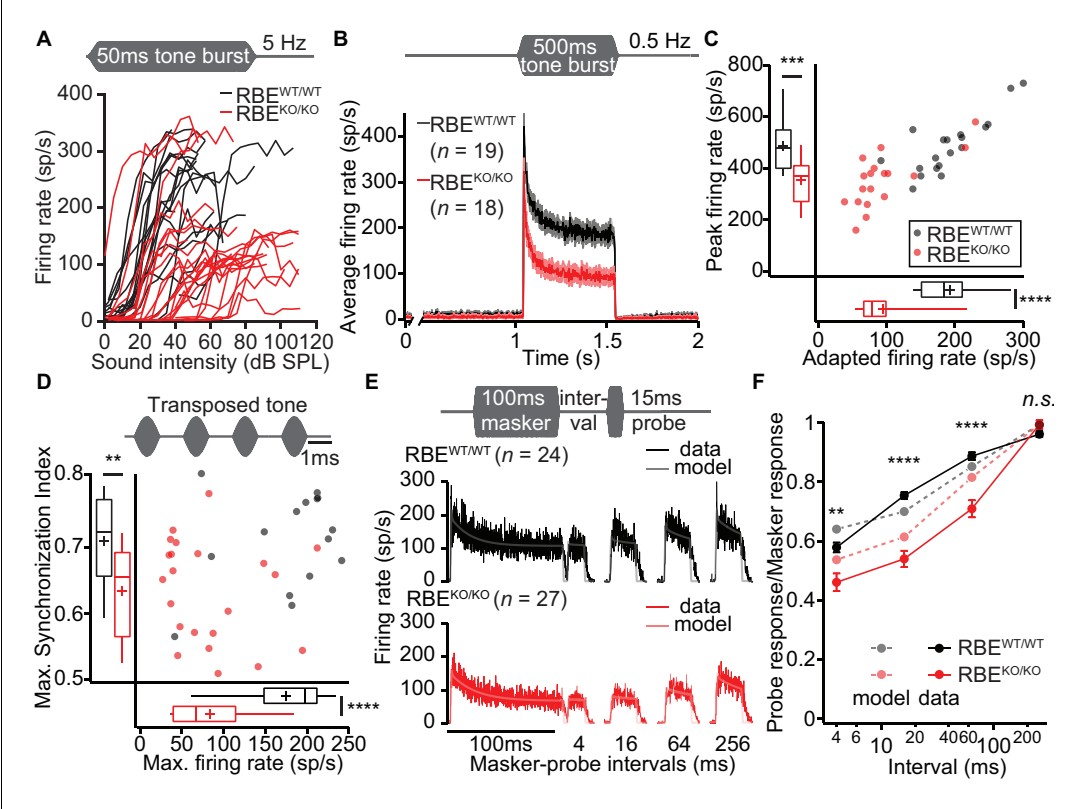

**Figure 11.** Lack of ribbons impairs vesicle replenishment in RBE$^{KO/KO}$ mice. (**A**) Intensity coding was largely preserved for suprathreshold sound stimulation: rate-level functions (average increase in spike rate with stimulus intensity) of SGNs in response to 50 ms tone bursts at C$_f$ at 5 Hz corroborated the notion of increased sound threshold but were otherwise comparable in RBE$^{KO/KO}$ ($n = 24$ SGNs, N = 8) and RBE$^{WT/WT}$ ($n = 19$ SGNs, N = 7) mice, for quantification see **Figure 11—figure supplement 1C**. (**B**) Average PSTH recorded in response to 50 × 500 ms tone bursts at C$_f$ and 30 dB above threshold at a stimulus rate of 0.5 Hz (bin width = 2 ms): the onset response to this stimulus was preserved better in RBE$^{KO/KO}$ SGNs ($n = 18$ SGNs, N = 7) as compared to higher stimulation rates (**Figure 9**), but it was still lower than in RBE$^{WT/WT}$ SGNs ($n = 19$ SGNs, N = 7), as well as the adapted firing rate. Mean (line) ± S.E.M. (shaded areas) are displayed. (**C**) Scatterplot of peak (highest 2 ms bin) and adapted (averaged 405–415 ms from response onset) firing rates: significantly reduced peak (p=0.0005, t-test) and adapted (p<0.0001, Mann-Whitney-Wilcoxon test) rates in RBE$^{KO/KO}$ SGNs. Box plots show 10, 25, 50, 75 and 90$^{th}$ percentiles and means are shown as crosses, as for (**D**). (**D**) Temporal precision and reliability of sound coding is impaired in RBE$^{KO/KO}$ SGNs. Synchronization of firing to stimulus and firing rates (reflecting spike probability) were reduced when probed with amplitude-modulated (transposed) tones (continuous stimulation with a carrier frequency at C$_f$ and at a modulation frequency of 500 Hz) (p=0.0043, t-test, for synchronization index, and p<0.0001, t-test, for firing rate, when comparing RBE$^{KO/KO}$ ($n = 22$ SGNs, N = 7) and RBE$^{WT/WT}$ ($n = 15$ SGNs, N = 6) SGNs). (**E**) Forward masking experiments were used to study presynaptic SV dynamics: a stimulus complex consisting of a 100 ms masker stimulus, a silent interval of 4/16/64/256 ms and a 15 ms probe (both at the characteristic frequency, 30 dB above threshold) was presented at 2 Hz. The averaged data after alignment of RBE$^{WT/WT}$ ($n = 24$ SGNs, N = 7, black) and RBE$^{KO/KO}$ ($n = 27$ SGNs, N = 8, red) SGNs responses are displayed as described in Materials and methods. On top of the data we present a fit of a biophysical model (light gray and pink lines) to the data used to study the SV dynamics at the AZ. The refilling and fusion rate constant during spontaneous and stimulated conditions as well as the number of occupied release sites are provided in **Table 2**. (**F**) Recovery of onset response showed as ratio of probe and masker response (number of spikes during first 10 ms, mean ± S.E.M), and prediction (dashed lines) derived from the model fit shown in (**E**). Recovery was slower in RBE$^{KO/KO}$ SGNs ($n = 27$ SGNs, N = 8) as compared to RBE$^{WT/WT}$ ($n = 24$ SGNs, N = 7) with significant differences in the ratio after 4 ms (p=0.0019, t-test), 16 ms (p<0.0001, Mann-Whitney-Wilcoxon test), and 64 ms masker-probe intervals (p<0.0001, Mann-Whitney-Wilcoxon test), but not after 256 ms (p=0.0835, t-test).

DOI: https://doi.org/10.7554/eLife.29275.019

The following figure supplement is available for figure 11:

**Figure supplement 1.** Rate-level functions and dynamic range remained unchanged in RBE$^{KO/KO}$.
DOI: https://doi.org/10.7554/eLife.29275.020

seen with shorter inter-stimulus interval (e.g. 200 ms, **Figure 9C**), likely reflecting more complete SV-replenishment (i.e. larger standing RRP) owing to the longer recovery interval (2 s vs. 200 ms). However, the adapted spike rate of RBE$^{KO/KO}$ SGNs was even more reduced than found with 50 ms tone bursts (to about half of that for RBE$^{WT/WT}$ SGNs, **Figure 11B,C**) highlighting the impaired SV

**Table 1.** Average double-exponential fitting results to peristimulus time histograms obtained by 50 ms tone bursts 30 dB above threshold at $C_f$ (200 ms inter-stimulus interval).

| | Fast time constant | Slow time constant | Amplitude fast component | Amplitude slow component | R |
|---|---|---|---|---|---|
| RBE$^{WT/WT}$ | 6.31 ± 0.77 | 95.63 ± 24.21 | 484.61 ± 104.76 | 58.42 ± 12.90 | 0.93 ± 0.00 |
| RBE$^{KO/KO}$ | 18.79 ± 6.91 | 101.66 ± 27.77 | 310.56 ± 73.60 | 64.92 ± 107.49 | 0.91 ± 0.01 |
| *p*-value | 0.0045 | 0.3580 | 0.0519 | 0.5475 | 0.2041 |

DOI: https://doi.org/10.7554/eLife.29275.018

replenishment during prolonged stimulation. Finally, we evaluated presynaptic vesicle pool dynamics by recording and modeling responses to forward masking protocols (*Harris and Dallos, 1979*), that are thought to reflect depletion and recovery of the RRP (*Figure 11E,F*; *Figure 11—figure supplement 1C*). We approximated the recovery from forward masking by single exponential fitting (*Figure 11—figure supplement 1C*) to provide an estimate of the kinetics of vesicle pool replenishment. The time constant of recovery was prolonged in RBE$^{KO/KO}$ SGNs (90.80 ± 8.66 ms, S. D. = 45.00 ms, n = 27 SGNs, N = 8 vs. 33.53 ± 5.74 ms, S.D. = 28.11 ms, in RBE$^{WT/WT}$ SGNs, *n* = 24 SGNs, N = 7, p<0.0001, Mann-Whitney-Wilcoxon test) indicating slowed RRP replenishment in the absence of the ribbon. We noted that SGNs showed considerably lower spontaneous and evoked rates during the forward masking paradigm compared to other stimulus protocols (compare *Figure 11B E*), likely due to enhanced RRP depletion with the more extended stimulation in this protocol.

Amplitude and waveform of the forward masking responses were consistent with a two-fold reduction of the number of contributing vesicular release sites of the RRP (N'$_{slot}$: contributing release sites during forward masking, N$_{slot}$: the contributing release sites for the same AZs during tone bursts at 5 Hz stimulation). The ratio N$_{slots}$/N'$_{slots}$ was estimated to be 2.2 for RBE$^{KO/KO}$ and 2.3 for RBE$^{WT/WT}$ from the drop in spontaneous and evoked SGN spiking rates. We used a previously developed biophysical model of RRP dynamics and spike generation (*Frank et al., 2010*; *Jung et al., 2015b*) to extract information on fusion and replenishment rate constants as well as the N$_{slot}$ by fitting the responses to tone bursts, that is PSTHs with 100, 200 and 2000 ms inter-stimulus interval, and also the forward masking spiking data across all recovery intervals (*Figure 11E*). Importantly, only the first 50 ms of the 500 ms stimulus response during the PSTH 2000 were included in the fit; any later adaptation processes were disregarded, as they were not accounted by the model's equations.

The results of model fitting suggested that during the forward masking only about half of all release sites (N$_{slots}$) were engaged in the response (N'$_{slots}$). Throughout, RBE$^{KO/KO}$ SGNs showed a lower fusion rate than the RBE$^{WT/WT}$ SGNs, reflecting the reduced onset response in RBE$^{KO/KO}$ SGNs. When more recovery time was allowed, that is in the 0.5 Hz tone burst and the forward masking, where recovery times from around 250 to 500 ms occured between the probe and subsequent masker, the onset response improved in RBE$^{KO/KO}$. Consequently, the estimated fusion rate almost reached the level of RBE$^{WT/WT}$ in the forward masking fits. With the scaling factors of approximately 2, the estimates for the number of release sites were consistent between tone bursts and forward

**Table 2.** Parameters for the biophysical model capturing the release dynamics during forward-masking and repetitive tone burst experiments

| | Forward masking | | All tone bursts – global fit | |
|---|---|---|---|---|
| | RBE$^{WT/WT}$ | RBE$^{KO/KO}$ | RBE$^{WT/WT}$ | RBE$^{KO/KO}$ |
| N$_{slots}$ | 8 | 6.5 | 15.6 | 13.7 |
| k$_{refill, stim}$ | 29.5 | 18.2 | 18.4 | 19.9 |
| k$_{refill, spont}$ | 13.3 | 7.1 | 5.59 | 2.71 |
| k$_{fusion, stim}$ | 32.2 | 31 | 82.5 | 45.1 |
| k$_{fusion, spont}$ | 0.6 | 0.9 | 0.63 | 0.54 |

DOI: https://doi.org/10.7554/eLife.29275.021

masking data, and in both cases only slightly smaller for the RBE$^{KO/KO}$ (*Table 2*). Taken together the fits from forward masking and tone bursts suggest that the total number of release sites (RRP) was only slightly reduced at ribbonless synapses of RBE$^{KO/KO}$ IHCs. However, a strong firing response at sound onset, that is release of a large standing RRP, required longer recovery indicating more efficient SV replenishment in the presence of ribbons, which is reflected in the larger refilling rate constants estimated by the model of RBE$^{WT/WT}$ synapses.

## Discussion

The role of the synaptic ribbon has remained a topic of intense research. Here, we studied the structure and function of IHC afferent synapses with SGNs in mice lacking RIBEYE, the core component of the synaptic ribbon. Morphologically, synapses of RBE$^{KO/KO}$ IHCs did not simply lack ribbons but instead appeared transformed to contacts where release from multiple AZ feeds into one postsynaptic bouton. Synaptic transmission was impaired at the ribbonless IHC synapses of RBE$^{KO/KO}$ mice. Spontaneous SGN firing was reduced, sound-evoked firing had higher sound thresholds, lower peak and adapted rates, recovered more slowly from adaptation and had a greater temporal jitter. Modelling of synaptic sound encoding corroborated the notion of reduced rates of SV fusion and replenishment at the ribbonless synapses of RBE$^{KO/KO}$ mice. Analysis of IHC function revealed a small depolarized shift in the operating range of the synaptic Ca$_V$1.3 Ca$^{2+}$-channels, which likely contributes to the reduced spontaneous and evoked firing rates and to the elevated sound thresholds of RBE$^{KO/KO}$ SGNs. $\Delta C_m$ recordings revealed a mild reduction of exocytosis but only for weaker depolarizations, which we primarily attribute to the depolarized shift of Ca$^{2+}$-channel activation. In summary, our data support a role of the ribbon in vesicle replenishment and Ca$^{2+}$-channel regulation at the AZs as required for synchronous activation of SGNs in normal hearing. However, our analysis of RBE$^{KO/KO}$ mice likely underestimated the role of the ribbon due to substantial compensation that is best illustrated by the striking transformation of AZ morphology and the mild ex vivo phenotype of IHC Ca$^{2+}$-influx and exocytosis.

### RIBEYE deletion transforms IHC synapses to 'conventional-like' presynaptic ultrastructure, where multiple ribbonless AZs collectively maintain large complements of Ca$^{2+}$-channels and SVs

Our work confirms the central role of RIBEYE for forming synaptic ribbons (*Schmitz et al., 2000*; *Magupalli et al., 2008*; *Maxeiner et al., 2016*). We did not observe structures reminiscent of 'ghost ribbons' reported for *ribeye* mutants in zebrafish neuromast hair cells (*Lv et al., 2016*) in IHCs of RBE$^{KO/KO}$ mice. These ghost ribbons were characterized as a halo of synaptic vesicles around a non-electron-dense area that resembled in size, though smaller, and shape to a synaptic ribbon. In zebrafish, two gene copies of *ribeye* (*ribeye a* and *b*) exist, making it harder to achieve a complete knockout (*Lv et al., 2016*; *Van Epps et al., 2004*). In keeping with this notion, Lv *et al.* found residual immunofluorescence of ribeye a in the double mutants. Hence, we speculate that residual RIBEYE, possibly together with other scaffold proteins such as piccolo, might have formed the observed electron-translucent SV-framed structures (*Lv et al., 2016*). In contrast, immunofluorescence, as well as electron microscopy, revealed the complete absence of RIBEYE and ribbons in IHCs of RBE$^{KO/KO}$ mice in our work and the companion study (Becker *et al.*), which is in agreement with findings in the RBE$^{KO/KO}$ mouse retina (*Maxeiner et al., 2016*).

IHC synapses normally employ a single ribbon-type AZ. But in the absence of RIBEYE, there were typically two or more ribbonless AZs, akin to multiple conventional AZs (*Figure 2*). These ribbonless 'conventional' AZs at RBE$^{KO/KO}$ IHC synapses consist mostly of roundish PDs, each with a cluster of SVs, of which approximately one third were directly adjacent to the plasma membrane (membrane-proximal: MP-SVs). Using electron tomography we found that about two-thirds of the MP-SVs were tethered to the AZ membrane, which was comparable to RBE$^{WT/WT}$ AZs (*Figure 3*). We speculate that SVs associated with the PD, but not facing the membrane (PDA-SVs), serve to replenish the release sites once tethered MP-SVs fused, and that the ribbonless PD more likely acts in long-range SV tethering to the AZ in analogy to what is considered for conventional AZs (*Cole et al., 2016*; *Fernández-Busnadiego et al., 2013*; *Siksou et al., 2007*). We assume that absence of RIBEYE does not alter SV size since electron tomography, which provides the most reliable estimation of SV size, did not reveal differences in SV diameter between RBE$^{KO/KO}$ and RBE$^{WT/WT}$ AZs, at least when

considering all SVs. The RBE$^{KO/KO}$ PDs, like in RBE$^{WT/WT}$, contained bassoon, Ca$_V$1.3, and RIM2, but lacked piccolino which is likely part of the ribbon in RBE$^{WT/WT}$ (*Figure 1*) (*Dick et al., 2001*; *Khimich et al., 2005*; *Limbach et al., 2011*; *Regus-Leidig et al., 2013*).

To some extent, the multi-AZ morphology is reminiscent of IHC synapses prior to synaptic maturation (*Huang et al., 2012*; *Sendin et al., 2007*; *Sobkowicz et al., 1982*; *Wong et al., 2014*). While we cannot rule out some sort of developmental delay of RIBEYE-deficient IHCs, we suspect that the morphological transformation into a multi-AZ morphology reflects a compensatory effort. Reasons for our interpretation include (i) the same morphological phenotype of RBE$^{KO/KO}$ IHCs at 8 months of age (*Figure 2*), (ii) the finding of highly regular PDs at RBE$^{KO/KO}$ IHCs synapses, which differs from less well-defined PDs at immature AZs (*Wong et al., 2014*), (iii) the typical continuous and large PSD of RBE$^{KO/KO}$ IHCs synapses (*Figure 2*, see also the accompanying paper by Becker *et al.*) as a characteristic of a mature synapse, rather than the several smaller PSD patches at developing IHC synapses (*Wong et al., 2014*), and (iv) the synaptically confined Ca$_V$1.3 Ca$^{2+}$-channel clusters, normal amplitude of IHC I$_{Ca}$ and mature amplitude of synaptic Ca$^{2+}$-signals, rather than massive extrasynaptic Ca$_V$1.3 abundance and larger whole-cell I$_{Ca}$ but smaller synaptic Ca$^{2+}$-signals in immature IHCs (*Wong et al., 2014*; *Zampini et al., 2010*). The multi-AZ morphology of the RBE$^{KO/KO}$ IHC synapses was also corroborated by high- and super-resolution microscopy of bassoon and Ca$_V$1.3 immunofluorescence (*Figure 4*). The organization in several smaller Ca$^{2+}$-channel clusters likely explains the broader spread of the presynaptic Ca$^{2+}$-signal at RBE$^{KO/KO}$ synapses (*Figure 6*). In contrast to bassoon mutant mice (*Frank et al., 2010*; *Jing et al., 2013*), the number of synaptic Ca$^{2+}$-channels was not reduced in RBE$^{KO/KO}$ mice as shown by normal amplitudes of whole-cell I$_{Ca}$ and synaptic Ca$^{2+}$-signals. Therefore, the loss of synaptic Ca$^{2+}$-channels from the bassoon-deficient ribbonless IHC synapses, indicates a role of bassoon in promoting Ca$^{2+}$-channel tethering at the AZ likely via interaction with RIM-binding protein (*Davydova et al., 2014*), which was previously shown to interact with Ca$_V$1.3 Ca$^{2+}$-channels (*Hibino et al., 2002*) and is required for establishing a normal Ca$^{2+}$-channel complement of the IHC AZ (*Krinner et al., 2017*).

Interestingly, we observed changes in Ca$^{2+}$-channel function in RBE$^{KO/KO}$ IHCs: the voltage-dependence of Ca$^{2+}$-channel activation was slightly, but significantly, shifted to more depolarized potentials both at the levels of whole-cell Ca$^{2+}$-current (V$_h$ +2 mV) and synaptic Ca$^{2+}$-influx at individual synapses (V$_h$ +5 mV) (at 5 mM [Ca$^{2+}$]$_e$, *Figure 5 and 6*). Similar as in this study, an enhanced inactivation (*Figure 5*) of I$_{Ca}$ was also found in bassoon-deficient IHCs, while their V$_h$ was actually mildly shifted in the opposite direction (−3 mV for imaging of synaptic Ca$^{2+}$) and unaltered at the level of the whole-cell I$_{Ca}$ (*Frank et al., 2010*). One potential reason for why the depolarized V$_h$-shift of the synaptic Ca$^{2+}$-influx was greater than that of the whole-cell Ca$^{2+}$-influx is the contribution of extrasynaptic Ca$^{2+}$-channels to the whole-cell Ca$^{2+}$-influx. They are thought to contribute approximately 30% of the Ca$^{2+}$-influx (*Brandt et al., 2005*) and are not regulated by RIBEYE/ribbon. In order to test whether the depolarized V$_h$-shift of synaptic Ca$^{2+}$ translates into changes in transmitter release, we recorded exocytic $\Delta C_m$ for different depolarization potentials. A small, but significant reduction of exocytosis for weak depolarizations in RBE$^{KO/KO}$ IHCs (*Figure 7*, seen also in the accompanying paper by Becker *et al.*) suggests that the V$_h$ shift is relevant for hair cell transmission (see also below). How enhanced I$_{Ca}$ inactivation might affect sound encoding is addressed by work on Ca$^{2+}$-binding proteins (CaBPs) that are thought to antagonize calmodulin's role in mediating I$_{Ca}$ inactivation (*Lee et al., 1999*; *Peterson et al., 1999*). Among the several CaBPs expressed in IHCs (*Cui et al., 2007*; *Picher et al., 2017*; *Schrauwen et al., 2012*; *Yang et al., 2006*), CaBP2 is defective in human genetic hearing loss *DFNB93* (*Picher et al., 2017*; *Schrauwen et al., 2012*) and required for hearing likely via inhibition of IHC I$_{Ca}$ inactivation (*Picher et al., 2017*). However, deletion of CaBP4 in mice caused only a very mild increase of I$_{Ca}$ inactivation similar to the one found here and did not alter auditory brainstem responses (*Cui et al., 2007*). Future studies need to address how RIBEYE/ribbons mechanistically regulate the function and spatial organization of Ca$^{2+}$-channels.

## What can the RIBEYE knock-out tell us about the function of the ribbon in sensory coding?

Over some decades, research on retinal photoreceptors and bipolar cells, on hair cells of the inner ear and the lateral lines, on electroreceptors as well as pineal cells, has aimed to elucidate the function(s) of the synaptic ribbon. Current hypotheses state that the ribbon functions in (i) replenishing

release sites ([*Bunt, 1971*; *Frank et al., 2010*; *von Gersdorff et al., 1996*; *Lenzi et al., 2002*; *Maxeiner et al., 2016*; *Snellman et al., 2011*; *Vaithianathan and Matthews, 2014*] for a deviating view see {*Jackman et al., 2009*}), potentially by facilitated diffusion of SVs on the ribbon surface towards the site of consumption (*Graydon et al., 2014*) and SV priming (*Grabner and Zenisek, 2013*; *Snellman et al., 2011*), (ii) establishing a large complement of vesicular release sites and Ca²⁺-channels at the active zone (*Frank et al., 2010*; *Khimich et al., 2005*), which remained hard to disentangle from potential function of bassoon (*Frank et al., 2010*; *Jing et al., 2013*), (iii) ensuring close spatial coupling of Ca²⁺-channels and vesicular release sites (*Maxeiner et al., 2016*) or enhancing presynaptic Ca²⁺-signals by limiting diffusional Ca²⁺-spread (*Graydon et al., 2011*), (iv) contributing to multivesicular release (*Graydon et al., 2011*; *Jing et al., 2013*; *Mehta et al., 2013*), and (v) contributing to SV reformation from endocytosed membranes (*Jung et al., 2015b*; *Khimich et al., 2005*; *Schwarz et al., 2011*). Clearly, SV-replenishment was impaired at the ribbon-less IHC synapses of RBE$^{KO/KO}$ mice. This is shown by slowed recovery from forward-masking and the use-dependent reduction of peak and adapted firing rates, which we further scrutinized by modeling. Therefore, our study supports a role of the ribbon in vesicle replenishment, which is also found in the accompanying paper by Becker *et al.*. Why RRP-recovery was not significantly altered when probed with pairs 20 ms long maximal Ca²⁺-influx by membrane capacitance measurements in IHCs (*Figure 7*) will need to be addressed in future studies, ideally using paired pre-and postsynaptic recordings of synaptic transmission with depolarizations of varying strength.

Each of the ribbon-manipulations employed to analyze its role has strengths, but also weaknesses, such as changes in other AZ proteins and long-term compensatory processes (e.g. bassoon deletion), complex manipulations (e.g. diurnal changes or hibernation: [*Hull et al., 2006*; *Mehta et al., 2013*; *Spiwoks-Becker et al., 2004*]) and photoablation (*Mehta et al., 2013*; *Snellman et al., 2011*). Genetic RIBEYE manipulations (*Lv et al., 2016*; *Maxeiner et al., 2016*; *Sheets et al., 2011*; *Van Epps et al., 2004*) have the greatest molecular specificity, but in some cases, were incomplete, and to some extent masked by compensation. In fact, our study of IHCs, unlike the situation for bipolar cell retinal ribbons (*Maxeiner et al., 2016*), suggests that some features of the IHC ribbon-type AZ can be very well replaced by a ribbonless multi-AZ morphology: the synaptic complement of Ca²⁺-channels and SVs, as well as exocytic $\Delta C_m$ elicited by strong depolarizations, were similar. Therefore, we likely underestimated the role of the ribbon in sound encoding in our present study.

For sound encoding at the afferent synapses between IHCs and SGNs, we observed some commonalities and differences with the bassoon mutants and RBE$^{KO/KO}$ mice, whereby the stronger phenotype of bassoon mutants suggests additive effects of bassoon and ribbon loss. Recordings from single SGNs indicate reduced peak and adapted release rates at the IHC synapses of ribbonless synapses, as well as impaired temporal precision of coding. High temporal precision is a hallmark of synaptic sound encoding (e.g. (*Köppl, 1997*). Reduced release rates or smaller EPSC sizes would increase the temporal jitter (*Buran et al., 2010*; *Li et al., 2014*; *Rutherford et al., 2012*; *Wittig and Parsons, 2008*). Reduced spike rates and increased jitter of release likely explain the reduced ABR wave I amplitude in both mutants. A striking difference from bassoon mutants, however, is that sound encoding in RBE$^{KO/KO}$ mice was impaired substantially, despite unaltered exocytic $\Delta C_m$ upon strong stimulation. We propose two mechanisms with likely additive effects to explain this surprising finding: i) the small depolarized shift of synaptic Ca²⁺-channels might contribute the lower spontaneous and evoked firing rates as well as higher sound thresholds of SGNs and ii) the reduced SV-replenishment might not suffice to balance the rate of consumption leading to a smaller standing RRP *in vivo*, while the arrest of exocytosis in the voltage-clamped IHCs for tens of seconds likely enables complete filling of the release sites (max. standing RRP). The changes in Ca²⁺-channel gating observed in IHCs of RBE$^{KO/KO}$ mice were unexpected, as so far, a direct or indirect interaction of RIBEYE and Ca²⁺-channels have not been described. Clearly, future studies, including studies on the potential regulation of Ca$_V$1.3 Ca²⁺-channels by RIBEYE and piccolino in heterologous expression systems, paired pre- and postsynaptic recordings, as well as further computational modeling, are required.

# Materials and methods

## Key resources table

| Reagent type (species) or resource | Designation | Source or reference | Identifiers | Additional information |
|---|---|---|---|---|
| Strain, strain background (*Mus musculus*) | Constitutive RIBEYE knockout, C57BL/6 background | PMID: 26929012 | | |
| Antibody | CtBP2 (mouse monoclonal) | BD Biosciences | 612044 | 1:200 |
| Antibody | PSD-95 (mouse monoclonal) | Sigma Aldrich | P246-100ul | 1:200 |
| Antibody | Bassoon SAP7f407 (mouse monoclonal) | Abcam | ab82958 | 1:200 |
| Antibody | Bassoon (guinea pig polyclonal) | Synaptic Systems | 141 004 | 1:500 |
| Antibody | RIM2 (rabbit polyclonal) | Synaptic Systems | 140 103 | 1:100 |
| Antibody | $Ca_v1.3$ (rabbit polyclonal) | Alomone Labs | ACC 005 | 1:75 or 1:100 |
| Antibody | Piccolino (rabbit polyclonal) | *Regus-Leidig et al. (2013)* | | 1:500 |
| Antibody | Parvalbumin α (guinea pig polyclonal) | Synaptic Systems | 195 004 | 1:1000 |
| Antibody | Calbindin 28 k (mouse monoclonal) | Swant | 07(F) | 1:500 |
| Antibody | Calretinin (rabbit polyclonal) | Swant | 1893–0114 | 1:1000 |
| Antibody | Alexa Fluor 488 conjugated anti-rabbit (goat polyclonal) | Invitrogen | A11008 | 1:200 |
| Antibody | Alexa Fluor 488 conjugated anti-guinea-pig (goat polyclonal) | Invitrogen | A11004 | 1:200 |
| Antibody | Alexa Fluor 568 conjugated anti-mouse (goat polyclonal) | Invitrogen | A11073 | 1:200 |
| Antibody | Alexa Fluor 647 conjugated anti-rabbit (donkey polyclonal) | Invitrogen | A31573 | 1:200 |
| Antibody | STAR580 conjugated anti-mouse (goat polyclonal) | Abberior | 2-0002-005-1 | 1:200 |
| Antibody | STAR580 conjugated anti-rabbit (goat polyclonal) | Abberior | 2-0012-005-8 | 1:200 |
| Antibody | STAR635p conjugated anti-mouse (goat polyclonal) | Abberior | 2-0002-007-5 | 1:200 |
| Antibody | STAR635p conjugated anti-rabbit (goat polyclonal) | Abberior | 2-0012-007-2 | 1:200 |
| Software, algorithm | Patchmaster or Pulse | http://www.heka.com/products/products_main.html#soft_pm | RRID:SCR_000034 | |
| Software, algorithm | IGOR Pro | http://www.wavemetrics.com/products/igorpro/igorpro.htm | RRID:SCR_000325 | |
| Software, algorithm | Patchers Power Tools | http://www3.mpibpc.mpg.de/groups/neher/index.php?page=software | RRID:SCR_001950 | |
| Software, algorithm | MATLAB | http://www.mathworks.com/products/matlab/ | RRID:SCR_001622 | |
| Software, algorithm | Gatan Microscopy Suite | http://www.gatan.com/products/tem-analysis/gatan-microscopy-suite-software | RRID:SCR_014492 | DigitalMicrograph scripting |
| Software, algorithm | Reconstruct | PMID: 15817063 | | |
| Software, algorithm | Serial-EM | PMID: 16182563 | | |
| Software, algorithm | 3dmod | PMID: 8742726 | | |
| Software, algorithm | IMOD | http://bio3d.colorado.edu/imod | RRID:SCR_003297 | |
| Software, algorithm | Genetic fit algorithm | *Sanchez del Rio and Pareschi, 2001* | doi: 10.1117/12.411624 | |
| Software, algorithm | Fiji | http://fiji.sc | RRID:SCR_002285 | |

*Continued on next page*

*Continued*

| Reagent type (species) or resource | Designation | Source or reference | Identifiers | Additional information |
|---|---|---|---|---|
| Software, algorithm | ImageJ | https://imagej.nih.gov/ij/ | RRID:SCR_003070 | |
| Software, algorithm | Imaris | http://www.bitplane.com/imaris/imaris | RRID:SCR_007370 | |
| Software, algorithm | Excel | microsoft.com/mac/excel | | |
| Software, algorithm | Origin | http://www.originlab.com/index.aspx?go=PRODUCTS/Origin | RRID:SCR_014212 | |
| Software, algorithm | GraphPad Prism | https://www.graphpad.com/scientific-software/prism/ | RRID:SCR_015807 | |
| Software, algorithm | Java Statistical Classes library | *Bertie, 2002* | | |

## Animals

Constitutive RIBEYE knockout mice (RBE$^{KO/KO}$ derived from *Ctbp2$^{tm1.2Sud}$* by Cre-recombination) were generated by Maxeiner and colleagues (*Maxeiner et al., 2016*) and were back-crossed to C57BL/6 for five generations (corresponding to a C57BL/6 background contribution of >95%). All experiments complied with national animal care guidelines and were approved by the University of Göttingen Board for Animal Welfare and the Animal Welfare Office of the State of Lower Saxony (permit number: 14–1391). The colony was maintained by mating heterozygous mice. Whenever possible, experiments were performed in parallel on mutant mice and their wildtype littermates. However, the experimental schedule did not always permit this and we occasionally used individual mice from the same colony but without littermate controls. Moreover, for some experiments giving rise to *Figure 6—figure supplement 1* and *Figure 7E,F,G,H* we also used C57Bl/6 wild-type mice and combined their results with those of wildtype littermate controls.

## Patch-clamp and confocal Ca$^{2+}$-imaging

The apical 2/3 turns of organs of Corti from P14 to P28 aged mice were freshly dissected in HEPES Hank's solution containing (in mM): 5.36 KCl, 141.7 NaCl, 10 HEPES, 0.5 MgSO$_4$.7H$_2$O, 1 MgCl$_2$-6H$_2$O, 5.6 D-glucose, and 3.4 L-glutamine (pH 7.2, ~300 mOsm). The basolateral membranes of the IHCs were exposed by carefully removing the surrounding cells with a suction pipette. All experiments were conducted at room temperature (20–25°C).

Perforated-patch-clamp recordings were performed as described previously (*Moser and Beutner, 2000*). The pipette solution contained (in mM): 130 Cs-gluconate, 10 tetraethylammonium (TEA)-Cl, 10 4-AP, 10 HEPES, 1 MgCl$_2$, as well as 300 µg/ml amphotericin B (pH 7.2, ~280 mOsm). The extracellular solution contained the following (in mM): 112 NaCl, 35 TEA-Cl, 2.8 KCl, 1 MgCl$_2$, 1 CsCl, 10 HEPES, 2 CaCl$_2$, and 11.1 D-glucose (pH 7.2, ~305 mOsm). External solution change was done by bath exchange through a perfusion system.

For live-imaging, the patch pipette solution contained (in mM): 111 Cs-glutamate, 1 MgCl$_2$, 1 CaCl$_2$, 10 EGTA, 13 TEA-Cl, 20 HEPES, 4 Mg-ATP, 0.3 Na-GTP and 1 L-Glutathione (pH 7.3, ~290 mOsm). To visualize the Ca$^{2+}$-hotspots and the ribbons, the Ca$^{2+}$-indicator Fluo-4FF penta-K$^+$ salt (0.8 mM, Life Technologies, Germany) and the TAMRA-conjugated CtBP2/RIBEYE-binding dimer peptide (10 µM, Biosynthan, Germany) were added to the intracellular solution. The extracellular solution contained the following (in mM): 2.8 KCl, 102 NaCl, 10 HEPES, 1 CsCl, 1 MgCl$_2$, 5 CaCl$_2$, 35 TEA-Cl, and 11.1 D-Glucose (pH 7.2, ~300 mOsm).

EPC-10 amplifiers controlled by Patchmaster or Pulse software (HEKA Elektronik, Germany) were used for the measurements. IHCs were held at −87 mV or −69 mV. All voltages were corrected for liquid junction potential offline (17 mV or 14 mV, depending on intra- and extracellular solutions used) and voltage-drops across the series resistance (R$_s$). Currents were leak corrected using a p/10 protocol in exocytosis experiments. Recordings were discarded when the leak current exceeded −55 pA, R$_s$ exceeded 30 MΩ (for perforated-patch) or 15 MΩ within 4 min after break-in (for ruptured-patch), or Ca$^{2+}$-current rundown exceeded 25%. All passive electrical properties of the patch-clamp recording experiments are detailed in *Supplementary file 2*.

Exocytosis was studied by measuring the membrane capacitance increments (ΔC$_m$) using the Lindau-Neher technique (*Lindau and Neher, 1988*). Cells were stimulated by step depolarizations of

different durations to $-17$ or $-14$ mV, or by 100 ms pulses to voltages ranging from $-53$ to $-37$ mV. A resting interval of 10–100 s between the stimuli was used. Each protocol was applied two to three times and only IHCs with reproducible exocytosis during the rounds were included. For display, traces were subjected to 1, 5 or 10 pass Binomial Smoothing using Igor Pro. Current-voltage relationships ('IVs') displayed in *Figure 5A* (ruptured-patch, 3-week-old mice) were obtained by applying 20 ms depolarizing step pulses of increasing voltage from $-82$ to 63 mV in 5 mV steps.

Ca$^{2+}$-imaging was performed with a spinning disk confocal scanner (CSU22, Yokogawa, Germany) mounted on an upright microscope (Axio Examiner, Zeiss, Germany) with 63x, 1.0 NA objective (W Plan-Apochromat, Zeiss). Images were acquired by a scientific CMOS camera (Neo, Andor, Germany). Ca$^{2+}$-indicator F4FF and TAMRA-conjugated peptide were excited by diode-pumped solid-state lasers with 491 nm and 561 nm wavelength, respectively (Cobolt AB). The spinning disk was set to 2000 rpm to synchronize with the 10 ms acquisition time of the camera.

Using a piezo positioner for the objective (Piezosystem, Germany), a scan of the entire cell was performed 4 min after breaking into the cell, taking sections each 0.5 µm at an exposure time of 0.5 s in the red (TAMRA-peptide) channel from the bottom to the top of the cell. In order to study the voltage-dependence of Ca$^{2+}$-indicator fluorescence increments at the synapses, the confocal scans were acquired every 0.5 µm from the bottom to the top ribbon in the RBE$^{WT/WT}$ mice. For the RBE$^{KO/KO}$ mice, the scanning was done from the bottom of the cell to +12 µm, which on average corresponds to the bottom of the nucleus. Ca$^{2+}$-currents were evoked by applying a voltage ramp stimulus from $-87$ to +63 mV during 150 ms (1 mV/ms) in each focal plane. Simultaneously, fluorescence measurements were made in the green channel (Fluo-4FF) with a frame rate of 100 Hz. In order to overcome the limitations of the frame rate and increase the voltage resolution of the fluorescent signal acquired, the voltage ramp protocol was applied twice, once shifted by 5 ms such that for any given frame during the second ramp the voltage was shifted by 5 mV compared to the first stimulus. Alternating planes were acquired to avoid photobleaching encountered with the consecutive plane acquisition.

## Immunohistochemistry, confocal and high resolution STED imaging

Apical turns of organs of Corti from 3-week-old mice were prepared for 'whole-mount imaging' as described in (*Ohn et al., 2016*). In brief, the samples were fixed either in formaldehyde (4%, 10 min on ice), or methanol (20 min at $-20°C$). Afterwards, the following primary antibodies were used: mouse anti-CtBP2 (1:200, BD Biosciences, Germany 612044), mouse anti-PSD-95 (1:200, Sigma Aldrich, Germany P246-100ul), mouse anti-bassoon SAP7f407 (1:200, Abcam, Germany, ab82958), guinea pig anti-bassoon (1:500, Synaptic Systems, Germany, 141 004), rabbit anti-RIM2 (1:100, Synaptic Systems 140 103), rabbit anti-Ca$_v$1.3 (1:75 or 1:100, Alomone Labs, Germany, ACC 005), rabbit anti-piccolino (1:500, kind gift of JH Brandstätter; see *Regus-Leidig et al., 2013*), guinea pig anti-parvalbumin α (1:1000, Synaptic Systems, 195 004), mouse anti-calbindin 28 k (1:500, Swant, Germany, 07(F)), and rabbit anti-calretinin (1:1000, Swant 1893–0114). Secondary antibodies used were Alexa Fluor 488 conjugated anti-rabbit, Alexa Fluor 488 conjugated anti-guinea-pig, Alexa Fluor 568 conjugated anti-mouse, and Alexa Fluor 647 conjugated anti-rabbit (1:200, Invitrogen, Germany, A11008, A11004, A11073, and A31573 respectively). For high resolution STED microscopy, STAR580 and STAR635p conjugated anti-rabbit and anti-mouse (1:200, Abberior, Germany, 2-0002-005-1, 2-0012-005-8, 2-0002-007-5, and 2-0012-007-2) have been used as secondary antibodies. Images were acquired using either a Leica SP5 with a 1.4 NA 63x oil immersion objective or an Abberior Instruments Expert Line STED microscope, with excitation lasers at 488, 561, and 633 nm and STED lasers at 595 nm, 1 W, and 775 nm, 1.2 W, using a 1.4 NA 100x oil immersion objective, either in confocal or in 2D-STED mode. Images were adjusted for brightness and contrast using ImageJ.

## Systems physiology: Auditory Brainstem Responses (ABR), Distortion Product Otoacoustic Emissions (DPOAE) and extracellular recordings from SGNs

ABR, DPOAE and extracellular recordings from single SGNs were performed essentially as described before (*Jing et al., 2013*; *Strenzke et al., 2016*). ABR and DPOAE recordings were performed on 6-week-old mice. For extracellular recordings from individual SGNs, 6 to 10 week-old mice were

anesthetized by i.p. injection of urethane (1.32 mg/kg), xylazine (5 mg/kg) and buprenorphine (0.1 mg/kg), a tracheostomy was performed and the mice were then placed in a stereotactic system. After partial removal of the occipital bone and cerebellum to expose the anteroventral cochlear nucleus (AVCN), a glass microelectrode was advanced through the posterior AVCN portion to reach the auditory nerve. Acoustic stimulation was provided by an open field Avisoft ScanSpeak Ultrasonic Speaker (Avisoft Bioacoustics, Germany), and 'putative' SGNs (auditory nerve fibers formed by the central SGN axons) were identified and distinguished from cochlear nucleus neurons based on their stereotactic position (>1.1 mm from the surface of the cochlear nucleus), spontaneous and noise-burst induced firing, peristimulus time histogram (PSTH), regularity of firing, and first spike latency. Recordings were performed using TDT system III hardware and an ELC-03XS amplifier (NPI Electronics, Germany), offline analysis using waveform-based spike detection using custom-written MATLAB software (*Source code 1*).

## Transmission electron microscopy
### Conventional embedding

Conventional embedding of organs of Corti was essentially performed as described previously (*Wong et al., 2014*). In brief, here P21, 6 weeks and 8 months old mice were used. The apical turn of organs of Corti were dissected in phosphate-buffer saline (PBS) and fixed for 1 hr on ice with 4% paraformaldehyde and 0.5% glutaraldehyde in PBS (pH 7.4). After an additional fixation overnight on ice with 2% glutaraldehyde in 0.1 M sodium cacodylate buffer (pH 7.2), samples were washed in 0.1 M sodium cacodylate buffer and placed in 1% osmium tetroxide (v/v in 0.1 M sodium cacodylate buffer) on ice for 1 hr. Next, samples were washed twice in 0.1 M sodium cacodylate buffer (10 min each, on ice) and further in distilled water and subsequently en bloc stained with 1% uranyl acetate (v/v in distilled water) for 1 hr on ice. Uranyl acetate treated samples were briefly washed three times in distilled water, dehydrated using a series of increasing ethanol concentration and finally embedded in epoxy resin (Agar 100 kit, Plano, Germany) and polymerized for 48 hr at 70°C. An Ultracut E microtome (Leica Microsystems, Germany) equipped with a 35° diamond knife (Diatome, Switzerland) was used to obtain ultrathin sections (70–75 nm) of the specimen. Sections were transferred to 1% formvar-coated (w/v in water-free chloroform) copper slot grids (ATHENE copper slot grids, 3.05 mm Ø, 1 mm x 2 mm; Plano, Germany) and subsequently stained with uranyl acetate replacement solution (UAR-EMS) (Science Services, Germany) and Reynold's lead citrate. The specimens were investigated at 80 kV with a JEM1011 transmission electron microscope (JEOL, Germany) and micrographs acquired at 10,000-x magnification using a Gatan Orius 1200A camera (Gatan, Germany, using the Digital Micrograph software package). Serial 3D reconstructions of ultrathin sections were generated with the program Reconstruct (*Fiala, 2005*).

## High-pressure freezing/freeze-substitution (HPF/FS) and electron tomography

High-pressure freezing, freeze-substitution followed by electron tomography were essentially performed as described previously (*Vogl et al., 2015*; *Jung et al., 2015a*). After freeze-substitution and embedding in epoxy resin (Agar 100 kit, Plano, Germany), 250 nm semithin sections for electron tomography were obtained on an Ultracut E ultramicrotome (Leica Microsystems, Germany) with a 35° diamond knife (Diatome, Switzerland). Sections were placed on 1% formvar-coated (w/v in water-free chloroform) copper 100 mesh grids (ATHENE, Plano, Germany, 3.05 mm Ø) and post-stained with UAR-EMS (Science Services, Germany) and Reynold's lead citrate.

For electron tomography, 10 nm gold particles (British Bio Cell/Plano, Germany) were applied to both sides of the stained grids. Single tilt series at 12,000-x magnification, mainly from −60 to +60° (if only fewer angles were possible, the tomograms were only accepted for quantification if the quality was sufficient) were acquired with an 1° increment at a JEM2100 (JEOL, Germany)) transmission electron microscope at 200 kV using the Serial-EM software (*Mastronarde, 2005*). The tomograms were generated using the IMOD package etomo and models were generated using 3dmod (*Kremer et al., 1996*).

## Data analysis

Live-imaging and IHC-patch-clamp data were analyzed using custom programs in Igor Pro 6.3 (Wavemetrics, Portland, OR, USA; *Source Code 2*). For analysis of IV-curves, the evoked $Ca^{2+}$-current was averaged from 5 to 10 ms after the start of the depolarization. The total $Ca^{2+}$-charge was estimated by taking the integral of the leak-subtracted current during the depolarization step. For most protocols, $\Delta C_m$ was estimated as the difference between the mean of $C_m$ 400 ms after and before the depolarization (the initial 60 ms after the end of depolarization were skipped). For paired pulse experiments, the calculation of the mean of $C_m$ before and after the depolarization was limited to the time remaining in the inter pulse interval after skipping (the initial 30 ms after the end of depolarization were skipped).

$\Delta F$ images were generated by subtracting the fluorescence intensity inside the cell at the resting state ($F_0$, an average of 10 frames) from the one at the depolarized state (an average of 6 frames during voltage ramp protocol). $\Delta F$ was calculated as the average of a $3 \times 3$ pixel square placed in the region showing the greatest intensity increase within the fluorescence hotspot. Maximal $\Delta F$ ($\Delta F_{max}$) was the average of 5 $\Delta F$ values obtained between $-17$ and $+8$ mV during the voltage ramp (around the peak $Ca^{2+}$-influx). Only AZs presenting a $\Delta F_{max}$ greater than the mean of the fluorescence intensity plus two standard deviations at rest were considered for further analysis. For analysis of the voltage dependence of synaptic $Ca^{2+}$-signals, raw traces were fitted to the following

$$F(V) = F_0 + \frac{f_v \cdot (V_r - V)}{1 + e^{\frac{(V_h - V)}{k}}} \tag{1}$$

where $f_v$ is the fluorescence-voltage-relationship $\Delta F / \Delta V$ obtained by linear fitting to the FV-curve in the range of 3 to 23 mV, $V_r$ the reversal potential of 65.6 mV, and $V$ the command voltage, in order to obtain $V_h$, the voltage of half-maximal activation, and $k$, the slope factor. The spatial extent of the synaptic $Ca^{2+}$-signals was estimated by fitting of a 2D Gaussian function to the fluorescent hotspot using a genetic fit algorithm (*Sanchez del Rio and Pareschi, 2001*) to obtain the full width at half maximum in the long and short axis. For each spot, the calculations were made at those confocal sections where the intensity of the spot was strongest.

Activation time constants of $Ca^{2+}$-currents at differing potentials were obtained by fitting to the first 5 ms of the current traces the following equation:

$$f(t) = y_0 + A \times (1 - e^{(\frac{-x}{\tau})})^2 \tag{2}$$

Confocal and STED immunofluorescence images were analyzed and z-projected with Fiji software and further analyzed using Igor Pro. For synapse counting, co-localized pre- and postsynaptic immunofluorescent spots were counted manually. The spatial extent of the line-shaped $Ca^{2+}$-channel clusters was estimated by fitting a 2D Gaussian function to the individual clusters in 2D STED images to obtain the full width at half maximum in the long and short axis. The areas of the PSD were calculated by the following formula: area = π x (Long Axis/2) (Short Axis/2). The semi-quantitative immunofluorescence analysis of the proteinaceous $Ca^{2+}$-buffers was performed by calculating the mean immunofluorescence intensity of a volume (40 (X) x 40 (Y) x 4 (Z) voxels or $2.8 \times 2.8 \times 2$ µm) below the nucleus and above the synapses. This and the count and intensity of the CtBP2 immunofluorescent spots have been analyzed in Imaris 7.6.5 with custom Matlab routines (*Source Code 4*).

For extracellular SGN recordings, PSTHs were calculated as average firing rates across 200 presentations of 50 ms or 500 ms tone bursts presented at 0.1 s/0.2 s or 2 s intervals, resp. (PSTH at 10/5 Hz and 0.5 Hz) at $C_f$, 30 dB above the threshold and binned at a width of 2 ms. Peak rate was determined as the largest bin of the PSTH in a time window 3–11 ms after stimulus onset. Adapted rate was averaged in a window spanning 35–45 ms or 405–415 ms after stimulus onset (for PSTH at 5 Hz and 0.5 Hz, respectively). Rate level functions were acquired using 50 ms tone bursts presented at $C_f$ and at 5 Hz. 25 repetitions for each stimulus intensity (5 dB steps) were recorded. Maximal steepness was calculated as the maximal increase in spike rate between two consecutive 5 dB increment steps. Dynamic range was calculated by using sigmoidal fits in the rate level functions as described in and measuring the range of sound pressure between 10% and 90% of maximal firing rate. For amplitude modulation analysis, synchronization index was calculated as described by

(*Goldberg and Brown, 1969*). Synchronization index estimation was only considered valid when at least 15 spikes occurred in a 3 s time window and the Rayleigh statistic was below 13.8.

For analysis of forward masking experiments, spike counts in a 10 ms interval starting from responses of both masker and probe onset were determined and presented as the ratio of probe and masker responses for at least 25 repetitions for every masker-probe interval from each unit. Exponential fitting to the plots of each individual SGN approximated the recovery kinetics.

Quantitative analysis of electron microscopy data was performed with ImageJ for conventional embedded samples and with IMOD for HPF/FS tomograms. According to the presence of ribbon-occupied and ribbonless synapses, we considered the following analysis criteria:

For ribbon-occupied synapses, membrane-proximal synaptic vesicles (MP-SVs, within a distance of ≤25 nm from the AZ membrane and ≤80 nm from the presynaptic density) and ribbon-associated synaptic vesicles (RA-SVs, first layer around the ribbon with a maximum distance of 80 nm from the vesicle membrane to the ribbon) were counted (*Figure 2J*, random sections analysis according to *Strenzke et al., 2016* and *Figure 3G* for tomograms according to 2D-random section analysis criteria). The tomogram analysis parameters were further modified, as used in *Jung et al., 2015a*. Here, the MP-SVs were defined as vesicles with ≤50 nm from the AZ membrane and with the shortest distance from the vesicle membrane to the presynaptic density of ≤100 nm, excluding RA-SVs (*Figure 3—figure supplement 1A*). For random sections, SV diameters were calculated by the averaged measurements of the horizontal and vertical axis. The ribbon size was measured in height and width, taking the longest axis of the ribbon excluding the PD, and the edges of the synaptic ribbon were traced manually using ImageJ. The length of the PD was measured along the AZ membrane (*Figure 2J*).

For **ribbonless** synapses, a presynaptic density-associated synaptic vesicle (PDA-SVs) pool was defined considering all clustered vesicles ≤80 nm around the PD that did not fulfill the criteria of a MP-SV (see above, also *Figure 2J* for random sections). The MP-SV pool, as well as the SV diameter and PD length, were analyzed as for the ribbon-occupied synapses. For tomograms, the PDA-SV pool was defined as the SVs in the first layer ≤ 80 nm to the PD, excluding the MP-SVs. The MP-SV pool criteria are the same as described in the previous paragraph (*Figure 3G*, according to 2D-random section and *Figure 3—figure supplement 1A*, according to *Jung et al., 2015a*). For tomograms, the according pools were further distinguished into tethered and non-tethered vesicles (*Figure 3G* and *Figure 3—figure supplement 1A*). All vesicles were annotated using a spherical 'scattered object' at its maximum projection in the tomogram, encompassing the outer leaflet of the vesicles. The vesicle radii were determined automatically (*Helmprobst et al., 2015*) with the program 'imodinfo option -p' of the IMOD software package (*Kremer et al., 1996*).

## Release site model of RRP release and replenishment

The coding of sound onset differs among the various SGNs in time due to different durations of the traveling wave, synaptic delays and conduction times. To obtain an average PSTH for modeling that is not smeared out due to such differences between units, the individual PSTHs were aligned beforehand by shifting their timing relative to each other. Onset detection was based on a change in spike statistics. For spontaneous activity, the 99.5 percentile of spike counts was determined. Next, the time at which response rises to twice this percentile was found. This is certainly a point within the sound response. Finally going back from this point, a drop back baseline activity, that is below the percentile was detected and used as onset time. Aligned PSTH from all units were averaged. This averaged PSTH from the forward masking data were fit with a model waveform using a genetic fit algorithm implemented in IGOR Pro (Wavemetrics, Lake Oswego, OR, USA). The purpose of the model is to give insight into the dynamics of SV cycling at the average IHC AZ. More specifically, the notion of $Ca^{2+}$-nanodomain-like control of RRP exocytosis (*Brandt et al., 2005*; *Graydon et al., 2011*; *Pangršič et al., 2015*; *Wong et al., 2014*), as well as the limited MP-SVs at the AZ (see *Figures 2* and *3*) motivates the notion of a limited, quasi-fixed number of available vesicular release sites or slots, $N_{slot}$, (*Frank et al., 2010*; *Wong et al., 2014*) that constitute the RRP. Each of these sites can be either empty or occupied by a release ready SV (whereby all filled slots constitute the 'standing' RRP) and at each time point, a release ready SV will fuse with a certain probability described by the fusion rate constant $k_{fus}$. Its value depends on the sound pressure level in a relation we assume to be linear within the dynamic range of the synapse/fiber. While the sound pressure level rises from silence to saturation $k_{fus}$ increases from $k_{fus, spont}$ to $k_{fus,stim}$. The refilling of empty

sites is described by a refill rate constant $k_{refill}$, which also depends on the sound intensity ($k_{refill, spont}$ to $k_{refill,stim}$).

The state of the release site was described by:

$$\frac{dN_{slot}^{filled}(t)}{dt} = k_{refill}(t)N_{slot} - N_{slot}^{filled}(t)\left(k_{fus}(t) + k_{refill}(t)\right) \tag{3}$$

Although this equation is formulated for SV fusion rates, a scaling factor f can be used to account for the fraction of fusion events that cannot successfully trigger an action potential (AP) despite sufficient neural excitability for example because of the too small size of the elicited excitatory post synaptic current. This factor effectively operates as if the number of release sites was scaled down. The scaled equation then gives a rate R of potentially supra-threshold EPSCs as the product of the number of occupied release sites, the fusion rate constant and the scaling factor f:

$$R(t) = k_{fus}(t).f.N_{slot}^{filled}(t) \tag{4}$$

The stationary solutions of *Equation 3* together with *Equation 4* determine steady state occupancy and steady state event rates:

$$N_{solt}^{filled}\Big|_{strady\,state}^{condition} = \frac{K_{refill}^{condition}}{K_{fusion}^{condition} + K_{refill}^{condition}}.N_{solt} \tag{5}$$

$$R\Big|_{strady\,state}^{condition} = \frac{K_{fusion}^{condition}.f.K_{refill}^{condition}}{K_{fusion}^{condition} + K_{refill}^{condition}}.N_{solt} \tag{6}$$

In this equation, 'condition' is either silence or saturating sound pressure level.

In order to connect the postsynaptic event rate of potentially supra-threshold EPSCs to the actual AP rate, refractoriness is considered as a combination of an absolute refractory period $t_{abs}$, during which the probability of an EPSC to trigger an AP is zero, with a relative refractory period during which this trigger probability returns to one with an exponential time course characterized by $\tau_{rel}$ (*Berry and Meister, 1998*). This description of refractoriness can be applied to the 'driving' EPSC rate R by means of a delayed differential equation. The equation is motivated by the concept of three possible states of the SGN: 'absolute refractory', 'relative refractory' or 'available' (fully excitable). At any point, the probability that the SGN turns from 'available' to 'refractory' is proportional to the rate R(t). The return back to 'available' happens 'delayed' by $t_{abs}$ and with a probability that is proportional to $1/\tau_{rel}$.

$$\frac{df_{avail}(t)}{dt} = \frac{f_{relref}(t)}{\tau_{rel}} - f_{avail}(t).f.R(t)$$
$$\frac{df_{relref}(t)}{dt} = f_{avail}(t - t_{abs}).f.R(t - t_{abs}) - \frac{f_{relref}(t)}{\tau_{rel}} \tag{7}$$

Together with *Equation 6* the stationary solution of this description of refractoriness connects the observable steady state rates during silence and stimulation to the rate constants $k_{refill}$ und $k_{fusion}$:

$$AP\_Rate\Big|_{strady\,state}^{condition} = \frac{K_{fusion}^{condition}.f.K_{refill}^{condition}}{K_{fusion}^{condition} + K_{refill}^{condition}}.N_{solt}.\frac{1}{1 + R\Big|_{strady\,state}^{condition}(t_{abs} + \tau_{rel})} \tag{8}$$

To go beyond the description of steady state event rates and to use the model for a parameterized description of the actual time course of experimentally observed PSTHs acquired during forward masking (*Figure 11E*), it is necessary to define the relation between the applied stimulus and the fusion and refill rate constants. For the experimental data presented here, the stimulus level was increased from silence to 30 dB above fiber threshold within a 4 ms ramp having a quarter of a $\sin^2$ shape. It was assumed that $k_{fusion}$ and $k_{refill}$ follow the stimulus increase simultaneously. The ordinary differential and delayed differential equations above were combined into a fit function. PSTHs (one masker followed by one probe) were averaged per genotype for each masker probe interval (4, 16, 64 and 256 ms) and were fitted in parallel with one parameter set. During experiments, trials were acquired in immediate succession without gaps. Therefore, the model implements cyclic boundary conditions for the occupancy of the slots. This model only captures the short term processes,

assuming that a set of experiments, for example forward masking trials quickly lead to a steady state. Slow adapting processes were not explicitly modeled. The observed drop of the apparent number of available slots in the forward masking experiments was described here as a change in the number of slots from $N_{slots}$ to a reduced capacity $N'_{slots}$ and for a given spiral ganglion neuron that was tested with tone bursts and forward masking, the ratio $N_{slots}/N'_{slots}$ could be estimated from the change in rates (see Results).

### Statistical analysis

The data were analyzed using *Matlab* (Mathworks), *Excel*, *Igor Pro 6* (Wavemetrics), *Origin 9.0* (Microcal Software), and *GraphPad Prism* (GraphPad Software). Averages were expressed as mean ± standard error of the mean (S.E.M.). For every dataset, the standard deviation (S.D.), number of replicates (*n*) and animals (N) were indicated. For *Figure 7*, $n_{min}$ corresponds to the minimum number of cells included in the analysis of each depolarization potential given that the number of cells for each potential differs. In order to compare two samples, data sets were tested for normal distribution (Jarque-Bera test, D'Agostino and Pearson omnibus normality test or the Shapiro-Wilk test) and equality of variances (F-test), followed by two-tailed unpaired Student's t-test, or, when data were not normally distributed and/or variance was unequal between samples, the unpaired two-tailed Mann-Whitney-Wilcoxon test was used. Cumulative distributions in *Figure 9A* were statistically compared using the Kolmogorov-Smirnov test. The ROUT method (Q = 0.1%) from *GraphPad Prism* was used to identify definitive outliers for *Figure 7H*.

For multiple comparisons, statistical significance was calculated by using one-way ANOVA test (two-way ANOVA in the case of ABR thresholds) followed by Tukey's test for normally distributed data or Kruskal-Wallis (K-W) test followed by non-parametric multiple comparisons test (NPMC) for non-normally distributed data.

For SV diameter quantifications in random sections, a custom-written routine using Java Statistical Classes library (JSC) (*Bertie, 2002*) was utilized for statistical analysis (*Source code 5*). Due to the tied ranks of SV diameter measurements obtained for random sections, their S.E.M. was used as a tolerance value for the usage of Kruskal-Wallis test as suggested by Bertie *et al*. in JSC library (*Bertie, 2002*), where two values were treated as equal if their difference was ≤ S.E.M.. The non-significant difference between samples is reported as *n.s.*, significant differences are reported as *p<0.05, **p<0.01, ***p<0.001, ****p<0.0001.

## Acknowledgements

We thank N Dietrich, T Goldak, S Gerke, C Senger-Freitag and I Herfort for expert technical assistance, Gerhard Hoch for software and analysis development, and Frank Schmitz for providing RBE[KO/WT] mice.

## Additional information

### Funding

| Funder | Grant reference number | Author |
| --- | --- | --- |
| Deutsche Forschungsgemeinschaft | Collaborative Research Center 889, projects A2, A6, and A7 | Carolin Wichmann Nicola Strenzke Tobias Moser |
| Deutsche Forschungsgemeinschaft | Tobias Moser Leibniz program MO 896/51 | Tobias Moser |
| Max-Planck-Gesellschaft | Max-Planck-Fellowship | Tobias Moser |
| Niedersächsisches Vorab | | Tobias Moser |

The funders had no role in study design, data collection and interpretation, or the decision to submit the work for publication.

## Author contributions

Philippe Jean, Data curation, Formal analysis, Investigation, Visualization, Writing—original draft, Writing—review and editing, PJ performed immunohistochemistry and confocal immunofluorescence microscopy, patch-clamp, Ca2+ imaging, computational modeling and prepared the manuscript; David Lopez de la Morena, Data curation, Formal analysis, Investigation, Visualization, Writing—original draft, Writing—review and editing, DL performed in vivo extracellular recordings from single SGNs and AVCN neurons, and contributed to the manuscript; Susann Michanski, Data curation, Formal analysis, Investigation, Visualization, Writing—original draft, Writing—review and editing, SM performed conventional embedding and electron microscopy of random and serial sections, contributed to high-pressure freezing and to the manuscript; Lina María Jaime Tobón, Data curation, Formal analysis, Investigation, Visualization, Writing—original draft, Writing—review and editing, LMJT performed patch-clamp capacitance measurements and contributed to the manuscript; Rituparna Chakrabarti, Data curation, Formal analysis, Investigation, Visualization, Writing—original draft, Writing—review and editing, RC performed high-pressure freezing, freeze-substitution and electron tomography, and contributed to the manuscript; Maria Magdalena Picher, Data curation, Formal analysis, Investigation, Visualization, Writing—original draft, MMP performed patch-clamp capacitance measurements and contributed to the manuscript; Jakob Neef, Data curation, Software, Formal analysis, Supervision, Investigation, Visualization, Methodology, Writing—original draft, JN performed immunohistochemistry, confocal and STED immunofluorescence microscopy, co-supervised PJ and LMJT, and contributed to the manuscript; SangYong Jung, Data curation, Formal analysis, Investigation, SYJ performed immunohistochemistry and confocal immunofluorescence microscopy; Mehmet Gültas, Software, Formal analysis, Methodology, MG contributed a statistical analysis of electron microscopy data; Stephan Maxeiner, Resources, Writing—original draft, SM contributed mutant mice and genetic expertise; Andreas Neef, Conceptualization, Software, Formal analysis, Supervision, Methodology, Writing—original draft, AN designed the study, performed and supervised computational modeling, and contributed to the manuscript; Carolin Wichmann, Conceptualization, Supervision, Visualization, Writing—original draft, Writing—review and editing, CW designed the study, supervised electron microscopy and tomography, and contributed to the manuscript; Nicola Strenzke, Conceptualization, Software, Supervision, Methodology, NS designed the study and supervised in vivo extracellular recordings from single SGNs; Chad Grabner, Conceptualization, Writing—original draft, CG designed the study and contributed to the manuscript; Tobias Moser, Conceptualization, Resources, Supervision, Funding acquisition, Validation, Visualization, Methodology, Writing—original draft, Project administration, Writing—review and editing, TM designed the study, prepared the manuscript and co-supervised PJ, DL and LMJT

## Author ORCIDs

Philippe Jean (iD) http://orcid.org/0000-0001-5325-1370
David Lopez de la Morena (iD) http://orcid.org/0000-0003-0835-2732
Susann Michanski (iD) https://orcid.org/0000-0001-5893-1981
Lina María Jaime Tobón (iD) http://orcid.org/0000-0002-6752-7750
Maria Magdalena Picher (iD) https://orcid.org/0000-0003-0722-3883
Jakob Neef (iD) http://orcid.org/0000-0002-4757-9385
Andreas Neef (iD) https://orcid.org/0000-0003-4445-7478
Carolin Wichmann (iD) http://orcid.org/0000-0001-8868-8716
Tobias Moser (iD) http://orcid.org/0000-0001-7145-0533

## Ethics

Animal experimentation: All experiments complied with national animal care guidelines and were approved by the University of Göttingen Board for Animal Welfare and the Animal Welfare Office of the State of Lower Saxony (permit number: 14-1391).

## Decision letter and Author response

Decision letter https://doi.org/10.7554/eLife.29275.031
Author response https://doi.org/10.7554/eLife.29275.032

## Additional files

### Supplementary files
• Source code 1. Matlab scripts for the waveform-based spike detection.
DOI: https://doi.org/10.7554/eLife.29275.022

• Source code 2. Live calcium imaging analysis.
DOI: https://doi.org/10.7554/eLife.29275.023

• Source Code 3. Routines for the analysis of whole-cell capacitance measurements
DOI: https://doi.org/10.7554/eLife.29275.024

• Source Code 4. Matlab scripts for the count andintensity of immunofluorescent spots.
DOI: https://doi.org/10.7554/eLife.29275.025

• Source code 5. Java routine for the SV diameter quantifications in random sections
DOI: https://doi.org/10.7554/eLife.29275.026

• Supplementary file 1. Modified tomogram analysis. The table depicts the modified tomogram analysis performed according to *Jung et al., 2015a* displaying the mean ± (S.E.M.), (S.D.), *p*-values, sample size and statistical tests used to compare RBE$^{WT/WT}$ and RBE$^{KO/KO}$. Refer to *Figure 3—figure supplement 1* for the graphs.
DOI: https://doi.org/10.7554/eLife.29275.027

• Supplementary file 2. Passive electrical properties of patch-clamp recording experiments.
The table shows the mean ± (S.E.M.) of the passive electrical properties across all ruptured and perforated patch-clamp recording experiments in RBE$^{WT/WT}$ and RBE$^{KO/KO}$ conditions.
DOI: https://doi.org/10.7554/eLife.29275.028

• Transparent reporting form
DOI: https://doi.org/10.7554/eLife.29275.029

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
