## [Decision Letter]

Thank you for submitting your article "The synaptic ribbon is critical for sound encoding at high rates and with temporal precision" for consideration by *eLife*. Your article has been reviewed by two peer reviewers, and the evaluation has been overseen by a Reviewing Editor and Andrew King as the Senior Editor. The reviewers have opted to remain anonymous.

This decision was reached by discussion between the Reviewing Editor and the reviewers, and the Reviewing Editor has assembled the following comments to help you prepare a revised submission. Both reviewers highlighted the fact that your study explores a major issue, the role of the ribbon in IHC synaptic exocytosis, and that the article is based on a large, high-quality data set and is very well written.

However, there is a major issue that needs to be resolved. Both reviewers felt that it was necessary to clarify the discordance between some of the results reported in the two jointly submitted manuscripts, which explore the same mouse Ribeye KO/KO mutant. These discrepancies include, in particular, the modifications to the Ca^2+^-currents of the KO mice. The authors of the two manuscripts are therefore requested to work together to iron out these discrepancies.

The work performed on the zebrafish Ribeye mutants also requires additional attention (Lv et al., Cell Reports 2016). This work shows that frameshift mutations in the gene encoding Ribeye lead to "ghost ribbons" in zebrafish neuromast hair cells. What evidence is there to suggest that the situation is different in the KO mouse mutant studied? This point should be addressed with care. The results obtained in the zebrafish mutants are generally insufficiently taken into account and discussed in this article, and should be expanded.

Reviewer #1:

There are at least 2-3 major discrepancies between this and the companion paper, despite the fact that both investigate the same RIBEYE KO mouse. First, there were no differences observed in ABR threshold as were reported in the companion paper. Second, there are significant differences in the biophysics of the Ca^2+^-current (i.e. V-half activation) between KO and WT mice that are not reported in the companion paper. This discrepancy could be due to the large variability in the data presented in both studies, possibly indicating some issues with IHC viability. Third, the DCm measurements were found to be significantly different between -43 and -53 mV in this paper, while the companion study only showed a difference at -40 mV and not at -45 mV.

Reviewer #2:

1) What was the average value of R_s_ for the data in Figure 5? If the R_s_ =20 MΩ and I(Ca) = 150 pA then the voltage-clamp error is 3 mV. So I am a bit skeptical by the small shifts reported to about 5 mV, although Figure 6 gives me more confidence in the conclusions. The labels in Figure 5 and Figure 6 and other Figures are much too small, so please increase the font size in the figures to the largest amount possible. Consider that the figures will be shrunk in size later on.

2) Figure 7 shows that strong stimulation reveals no difference between RBE-KO-mice and WT-mice for IHC exocytosis, but Figure 8 shows that ABR wave I amplitude is much smaller at 80 dB (a strong stimulus). This seems like a contradiction. What is the explanation? Perhaps a lack of highly synchronous EPSPs that trigger spikes in a phase-locked manner (variance in FSL; Figure 9) and an increase in spike failures due to smaller evoked EPSPs in the RBE-KO-mice? Some more discussion and explanation would be helpful.

3) Figure 11 shows a highly significant difference in the vector strength. The authors should also mention here that larger EPSCs are better phase-locked than smaller EPSCs (see Li et al., Neuron, 2014; see their Figure 8). So one would also expect lower vector strength and reduced phase locking in the KO mice to be related to a smaller frequency of large EPSCs. Indeed, this is what the companion paper of Becker et al. (Ricci group) seems to find. So it seems appropriate to mention this previous paper while discussing Figure 11.

4) Show the Bassoon-KO mice data also on Figure 9 (using a different color or symbol).

5) In contradiction to this paper, the companion paper of Becker et al. found significant changes in the ABR thresholds and no change in the Ca^2+^-current properties. This should be discussed briefly. Maybe the methods are different or the number of animals studied? German mice and American mice should behave the same, right?

---

## [Author Response]

[…] However, there is a major issue that needs to be resolved. Both reviewers felt that it was necessary to clarify the discordance between some of the results reported in the two jointly submitted manuscripts, which explore the same mouse Ribeye KO/KO mutant. These discrepancies include, in particular, the modifications to the Ca^2+^-currents of the KO mice. The authors of the two manuscripts are therefore requested to work together to iron out these discrepancies.The work performed on the zebrafish Ribeye mutants also requires additional attention (Lv et al., Cell Reports 2016). This work shows that frameshift mutations in the gene encoding Ribeye lead to "ghost ribbons" in zebrafish neuromast hair cells. What evidence is there to suggest that the situation is different in the KO mouse mutant studied? This point should be addressed with care. The results obtained in the zebrafish mutants are generally insufficiently taken into account and discussed in this article, and should be expanded.

We thank the reviewers for the appreciation of our work and the valuable comments that helped us to further improve our manuscript. We have performed additional experiments and analysis to address the reviewers’ comments, which included more electron tomography of synaptic structure further consolidating our findings, immunohistochemistry for studying the expression of Ca^2+^-binding proteins which was unaltered in the ribbon-deficient hair cells, and further patch-clamp experiments that e.g. corroborated the finding of reduced hair cell exocytosis for weak depolarizations, which was also found by the authors of the companion paper (Becker et al.). Together with our colleagues, we have worked on clarifying potential reasons for differences, generating more synergism by additional cross-referencing and have further harmonized the interpretation.

We have also jointly reflected the editorial comment asking us to better address the differences of the present studies and the work by the Zenisek group on zebrafish (Lv et al., 2016). As you know we had already cited this work in the original submittal, but at the request of the editor we have now added more reference to the work in the Introduction and Discussion part. Clearly, unlike the findings of Lv et al., 2016, we have neither seen any RIBEYE signal in immunofluorescence nor any structure compatible with their “ghost ribbon” phenomenon in any of our electron micrographs or tomograms in hair cells of the RIBEYE knock-out mice. We presume that, different from the complete deletion of A domain in mice (Maxeiner et al., 2016), the abolition of the duplicated *ribeye* genes in zebrafish using frameshift mutations was incomplete, leading to these above findings. Since this leaves some uncertainty on the interpretation of the functional phenotype, we would like to refrain from more in-depth comparison to the zebrafish findings. A paragraph addressing these differences has been added in the Discussion. Beyond this we focus on comparing to the companion study in the ear and previous results on the eye (Maxeiner et al., 2016) of RIBEYE knock-out mice.

Reviewer #1:There are at least 2-3 major discrepancies between this and the companion paper, despite the fact that both investigate the same RIBEYE KO mouse. First, there were no differences observed in ABR threshold as were reported in the companion paper.

The mean ABR thresholds of RBE^KO/KO^ mice were approximately 10dB higher across all frequencies than those of RBE^WT/WT^ mice in both studies. While this difference reached significance based on comparing 28 RBE^KO/KO^ mice and 22 RBE^WT/WT^ mice, it did not in our study with a lower n = 10 for all three genotypes. In response to the reviewer’s comment we also performed a 2-way ANOVA taking into consideration only the WT and KO data, but the difference did not reach significance either.

So, despite minor technical differences (age: 6 weeks in our case, 3 weeks in their case; stimulus onset ramp: 1ms cos2 ramp in our case, 0.5ms cos2 in their case; threshold definition: visual inspection with 10 dB precision in our case, wave I peak measurement and interpolation to find threshold five SD above noise floor in their case) between our studies, we both arrived at the same magnitude of threshold increase, which however due to the lower n in our case did not reach significance.

However, just as our colleagues, we did find a decreased amplitude of the first ABR wave which reflects the compound action potential of the spiral ganglion (Figure 8). Moreover, our study also revealed an increased threshold of the individual spiral ganglion neurons (Figure 11) in RBE^KO/KO^ mice. Given normal cochlear amplification, signified by the otoacoustic emissions, these findings reveal an impairment of synaptic sound encoding at the ribbonless IHC synapses of RBE^KO/KO^ mice.

In response to the reviewers comment we have now further commented on this in the manuscript:

“First, we recorded auditory brainstem responses (ABR) and found a significant reduction in the amplitude of wave I that reflects the SGN compound action potential (1.14 ± 0.13 µV, S.D. = 0.38 µV, N = 10 for RBE^KO/KO^vs. 3.30 ± 0.51 µV, S.D. = 1.54 µV, N = 10 for RBE^WT/WT^, *p* = 0.0007, NPMC test) […] We found a non-significant trend of ABR threshold to be increased across all frequencies in RBE^KO/KO^ mice (Figure 8, significant increase in ABR in the companion paper Becker *et al.* based on a larger sample, n = 28RBE^KO/KO^ mice vs. 22 RBE^WT/WT^ mice).”

Second, there are significant differences in the biophysics of the Ca^2+^-current (i.e. V-half activation) between KO and WT mice that are not reported in the companion paper. This discrepancy could be due to the large variability in the data presented in both studies, possibly indicating some issues with IHC viability.

A small but significant depolarized shift of the voltage at half-maximal Ca^2+^-activation (V_half_) *RBE^KO/KO^* IHC has been found both at the whole-cell level (approximately + 2 mV) using ruptured patch-recordings and at the single synapse level (approximately + 5 mV) by Ca^2+^-imaging. These data were recorded at an elevated [Ca^2+^]_e_ of 5 mM to improve the signal to noise ratio of imaging Ca^2+^-signals at single active zones. This approach in conjunction with another mouse mutant (*Gipc3)* had recently indicated that small changes in the operating range of Ca^2+^-channels can affect spontaneous firing and sound encoding of spiral ganglion neurons substantially (Ohn et al., 2016).

In response to the reviewer’s comment, we revisited additional recordings at 2 mM [Ca^2+^]_e_, but the V_half_ difference between RBE^KO/KO^ and RBE^WT/WT^ IHCs did not reach statistical significance there. So the larger signal in [Ca^2+^]_e_ of 5 mM might have helped to reveal the reported small V_half_ difference. A lower [Ca^2+^]_e_ of 2.8 mM was used by the authors of the companion paper, whose experiments also differed in the age of the mice used for patch-clamp (p10-13, hence younger age checked from around the onset of hearing: our data of Figure 5 and Figure 6 are from p21-28 mice). It is tempting to speculate that these differences led to the different observations. Nonetheless, we strongly believe that the V_half_ difference between RBE^KO/KO^ and RBE^WT/WT^ IHCs is relevant and further underline this by the finding of significantly reduced exocytic of ΔC_m_ in the KO for depolarizations to -45, -43 and -41 mV (see below).

Third, the DCm measurements were found to be significantly different between -43 and -53 mV in this paper, while the companion study only showed a difference at -40 mV and not at -45 mV.

We think that the basic observation is the same for both studies: a reduction of exocytic ΔC_m_ for weak depolarizations. The minor discrepancy likely originated from binning the responses to several voltages. Therefore, we have now done more experiments and performed the statistical analysis for the individual voltages. We found significantly reduced exocytosis in the KO for depolarizations to -45, -43 and -41 mV (significant at-41 mV only if removing outlier). This result is in accordance with the companion paper where the significant difference is at -44mV.

Reviewer #2:1) What was the average value of R_s_ for the data in Figure 5? If the R_s_ =20 MΩ and I(Ca) = 150 pA then the voltage-clamp error is 3 mV. So I am a bit skeptical by the small shifts reported to about 5 mV, although Figure 6 gives me more confidence in the conclusions. The labels in Figure 5 and Figure 6 and other Figures are much too small, so please increase the font size in the figures to the largest amount possible. Consider that the figures will be shrunk in size later on.

We do understand the concern and happily report the inclusion of the passive electrical properties of the patch-clamp recordings as Supplementary file 2 referred to from the Materials and methods section. In brief, the series resistance of recordings in Figure 5 was 10.5 MΩ for RBE^KO/KO^IHCs and 11.3 MΩ for RBE^WT/WT^IHCs, for Figure 6 (Ca^2+^-imaging) it was 14MΩ for RBE^KO/KO^IHCs and 13.8 MΩ for RBE^WT/WT^IHCs. Given the values and lack of systematic differences, we consider it unlikely that a technical reason explains the difference in the V_half_ of Ca^2+^-current between IHCs and active zones of both genotypes.

In response to the reviewer’s comment we have increased the font size of the figure labels throughout.

2) Figure 7 shows that strong stimulation reveals no difference between RBE-KO-mice and WT-mice for IHC exocytosis, but Figure 8 shows that ABR wave I amplitude is much smaller at 80 dB (a strong stimulus). This seems like a contradiction. What is the explanation? Perhaps a lack of highly synchronous EPSPs that trigger spikes in a phase-locked manner (variance in FSL; Figure 9) and an increase in spike failures due to smaller evoked EPSPs in the RBE-KO-mice? Some more discussion and explanation would be helpful.

Thanks for this insightful comment. We have further enhanced the discussion of temporal precision and the comparison between in vitroand in vivorecordings. We do think that it is hard if not impossible to equate the stimuli delivered in the patch-clamp to the sounds applied in vivo. Given the strong potassium conductances of inner hair cells (BK, Kv and KCNQ), we think that the receptor potential in vivois unlikely to reach the voltages we impose to elicit maximal voltage-gated Ca^2+^-influx (-17 mV) and this is the reason why we used “near-physiological” depolarizations in an additional set of recordings (Figure 7). However, due to the low signal to noise in membrane capacitance measurements (only some 10 vesicles (each contributing only a few tens of 10^-18^ Farad) fused at each of the dozen active zones in whole-cell recordings with approximately 10^-11^ Farad resting capacitance of the hair cell) we needed to use relatively long depolarizations (100 ms) and cannot well resolve RRP exocytosis.

Other major differences between our in vitroand in vivorecordings include: i) temperature (room temperature for in vitro and physiological temperature in vivo) and ii) likely the physiological state after dissection without oxygen supply from blood perfusion etc. This is why we and others are becoming increasingly interested to adjust the conditions in the patch-clamp to the in vivo conditions as closely as possible. However, at this point and for the scope of the present manuscript, we do put most emphasis on the analysis of spiral ganglion firing in vivoas it is best reporting the role of the ribbon in sound encoding. The results of membrane capacitance measurements obtained with mild depolarizations suggest at least some correspondence to the in vivofindings and offer one potential explanation for the discrepancy to the in vivofindings when studying exocytosis with maximal stimulation in vitro(discussed in the last paragraph of Discussion section).

3) Figure 11 shows a highly significant difference in the vector strength. The authors should also mention here that larger EPSCs are better phase-locked than smaller EPSCs (see Li et al., Neuron, 2014; see their Figure 8). So one would also expect lower vector strength and reduced phase locking in the KO mice to be related to a smaller frequency of large EPSCs. Indeed, this is what the companion paper of Becker et al. (Ricci group) seems to find. So it seems appropriate to mention this previous paper while discussing Figure 11.

We have now included the notion that larger EPSCs enable lower jitter of postsynaptic firing and also quote the nice Li et al., Neuron 2014 paper in the Discussion:

“High temporal precision is a hallmark of synaptic sound encoding (e.g. Köppl, 1997). Reduced release rates or smaller EPSC sizes would increase the temporal jitter (Buran et al., 2010; Li et al., 2014; Rutherford et al., 2012; Wittig and Parsons, 2008). Reduced spike rates and increased jitter of release likely explain the reduced ABR wave I amplitude in both mutants.”

However, we would like to refrain from over interpreting the findings of our colleagues since there was no significant reduction in the median EPSC amplitude in the companion paper.

4) Show the Bassoon-KO mice data also on Figure 9 (using a different color or symbol).

Done.

5) In contradiction to this paper, the companion paper of Becker et al. found significant changes in the ABR thresholds and no change in the Ca^2+^-current properties. This should be discussed briefly. Maybe the methods are different or the number of animals studied? German mice and American mice should behave the same, right?

Done. The mean ABR thresholds of RBE^KO/KO^ mice were approximately 10dB higher across all frequencies than those of RBE^WT/WT^ mice in both studies. While this difference reached significance based on comparing 28 RBE^KO/KO^ mice and 22 RBE^WT/WT^ mice, it did not in our study with a lower n = 10 for all three genotypes. In response to the reviewer’s comment we also performed a 2-way ANOVA taking into consideration only the WT and KO data, but the difference did not reach significance either.

So, despite minor technical differences (age: 6 weeks in our case, 3 weeks in their case; stimulus onset ramp: 1ms cos2 ramp in our case, 0.5ms cos2 in their case; threshold definition: visual inspection with 10 dB precision in our case, wave I peak measurement and interpolation to find threshold five SD above noise floor in their case) between our studies, we both arrived at the same magnitude of threshold increase, which however due to the lower n in our case did not reach significance.

However, just as our colleagues, we did find a decreased amplitude of the first ABR wave which reflects the compound action potential of the spiral ganglion (Figure 8). Moreover, our study also revealed an increased threshold of the individual spiral ganglion neurons (Figure 11) in RBE^KO/KO^ mice. Given normal cochlear amplification, signified by the otoacoustic emissions, these findings reveal an impairment of synaptic sound encoding at the ribbonless IHC synapses of RBE^KO/KO^ mice.

In response to the reviewers comment we have now further commented on this in the manuscript:

“First, we recorded auditory brainstem responses (ABR) and found a significant reduction in the amplitude of wave I that reflects the SGN compound action potential (1.14 ± 0.13 µV, S.D. = 0.38 µV, N = 10 for RBE^KO/KO^vs. 3.30 ± 0.51 µV, S.D. = 1.54 µV, N = 10 for RBE^WT/WT^, *p* = 0.0007, NPMC test)…We found a non-significant trend of ABR threshold to be increased across all frequencies in RBE^KO/KO^ mice (Figure 8, refer to significant increase in ABR in the companion paper Becker *et al.* based on a larger sample, n = 28RBE^KO/KO^ mice vs. 22 RBE^WT/WT^ mice).”

A small but significant depolarized shift of the voltage at half-maximal Ca^2+^-activation (V_half_)RBE^KO/KO^ IHC has been found both at the whole-cell level (approximately + 2 mV) using ruptured patch-recordings and at the single synapse level (approximately + 5 mV) by Ca^2+^-imaging. These data were recorded at an elevated [Ca^2+^]_e_ of 5 mM to improve the signal to noise ratio of imaging Ca^2+^-signals at single active zones. This approach in conjunction with another mouse mutant (*Gipc3)* had recently indicated that small changes in the operating range of Ca^2+^-channels can affect spontaneous firing and sound encoding of spiral ganglion neurons substantially (Ohn *et al.*, 2016).

In response to the reviewer’s comment, we revisited additional recordings at 2 mM [Ca^2+^]_e_, but the V_half_ difference between RBE^KO/KO^ and RBE^WT/WT^ IHCs did not reach statistical significance there. So the larger signal in [Ca^2+^]_e_ of 5 mM might have helped to reveal the reported small V_half_ difference. A lower [Ca^2+^]_e_ of 2.8 mM was used by the authors of the companion paper, whose experiments also differed in the age of the mice used for patch-clamp (p10-13, hence younger age checked from around the onset of hearing: our data of Figure 5 and Figure 6 are from p21-28 mice). It is tempting to speculate that these differences led to the different observations. Nonetheless, we strongly believe that the V_half_ difference between RBE^KO/KO^ and RBE^WT/WT^ IHCs is relevant and further underline this by the finding of significantly reduced exocytic of ΔC_m_ in the KO for depolarizations to -45, -43 and -41 mV.